# The transcription factor ATF3 switches cell death from apoptosis to necroptosis in hepatic steatosis in male mice

Yuka Inaba [1,2], Emi Hashiuchi[2], Hitoshi Watanabe [1], Kumi Kimura[3], Yu Oshima[3], Kohsuke Tsuchiya [4], Shin Murai [5], Chiaki Takahashi [6], Michihiro Matsumoto[7], Shigetaka Kitajima[8], Yasuhiko Yamamoto [3], Masao Honda [9,10], Shun-ichiro Asahara[11], Kim Ravnskjaer [12,13], Shin-ichi Horike [14], Shuichi Kaneko[9], Masato Kasuga[15], Hiroyasu Nakano [5], Kenichi Harada[16] & Hiroshi Inoue [1,2] ✉

Hepatocellular death increases with hepatic steatosis aggravation, although its regulation remains unclear. Here we show that hepatic steatosis aggravation shifts the hepatocellular death mode from apoptosis to necroptosis, causing increased hepatocellular death. Our results reveal that the transcription factor ATF3 acts as a master regulator in this shift by inducing expression of RIPK3, a regulator of necroptosis. In severe hepatic steatosis, after partial hepatectomy, hepatic ATF3-deficient or -overexpressing mice display decreased or increased RIPK3 expression and necroptosis, respectively. In cultured hepatocytes, ATF3 changes TNFα-dependent cell death mode from apoptosis to necroptosis, as revealed by live-cell imaging. In non-alcoholic steatohepatitis (NASH) mice, hepatic ATF3 deficiency suppresses RIPK3 expression and hepatocellular death. In human NASH, hepatocellular damage is correlated with the frequency of hepatocytes expressing ATF3 or RIPK3, which overlap frequently. ATF3-dependent RIPK3 induction, causing a modal shift of hepatocellular death, can be a therapeutic target for steatosis-induced liver damage, including NASH.

Hepatocellular death contributes to liver homoeostasis by eliminating and replacing impaired hepatocytes[1]. While little hepatocellular death occurs under normal conditions due to low turnover in the healthy liver, hepatocellular death plays an important role as a trigger of liver regeneration, as seen after partial hepatectomy or acute liver injury[1–3]. However, hepatic steatosis increases hepatocellular death during liver regeneration, exacerbating both acute and chronic liver damage[4–7]. Indeed, in steatotic liver, the elevated hepatocellular death cancels out the regeneration and causes prolonged liver dysfunction after partial hepatectomy or acute liver injury[8–10]. Furthermore, increased hepatocellular death and the resulting impaired regeneration are associated with the progression of non-alcoholic steatohepatitis (NASH), a

chronic liver disease with inflammation triggered by fat accumulation in hepatocytes[11–13].

The severity of the hepatic steatosis alters the role of hepatocellular death under pathological liver damage conditions. In mild steatosis (<30% of hepatocytes affected), the elevated hepatocellular death offsets the hepatocyte proliferation during fatty liver regeneration[1,3,4]. However, in severe steatosis (<60% of hepatocytes affected), the hepatocellular death triggers an inflammatory response, which results in a vicious cycle leading to additional cell death and exacerbation of liver damage[9].

The type of hepatocellular death in hepatic steatosis determines whether the death offsets proliferation or induces further cell death.

Apoptosis and non-apoptotic lytic cell death are two types of hepatocellular death, and the latter is induced in severe steatosis and responsible for the exacerbation of liver damage[1,4]. Indeed, during liver regeneration after partial hepatectomy, apoptosis is induced in mild hepatic steatosis, but lytic cell death is additionally evoked in severe hepatic steatosis[9]. Apoptosis is a non-inflammatory cell death that results in nuclear and cellular shrinkage while maintaining the integrity of the cell membrane, with minimal effects on the surrounding tissue[4,7]. In contrast, lytic cell death is characterised by cell membrane disruption and cell rupture, which elicits a strong inflammatory response and induces further cell death[4,7,14]. However, it remains unclear how the induction of these different types of cell death depend on hepatic steatosis severity.

While uncontrollable passive necrosis has been considered to be the mode of lytic cell death in severe hepatic steatosis, it has emerged that multiple modes of hepatocellular death co-exist side by side in hepatic steatosis[1]. Among the modes of lytic cell death, necroptosis is reported to contribute to the hepatocellular death in severe hepatic steatosis[4,7]. Necroptosis, which exhibits similar morphological features to necrosis, is controlled by receptor interacting protein kinase-3 (RIPK3)[7]. It is triggered by the sequential phosphorylation of RIPK3 and mixed lineage kinase domain-like (MLKL), which disrupts the cell membrane[7]. RIPK3 deficiency attenuates acetaminophen-induced acute hepatic injury[15] and NASH induced by methionine/choline-deficient diet (MCD) feeding[16,17]. However, the role of necroptosis in hepatic steatosis remains controversial because RIPK3 expression is too low to induce necroptosis in the healthy or mild steatotic liver. Ferroptosis, another mode of lytic cell death that is characterised by iron-dependent and lipid peroxidation-mediated cell death, may play a role in the pathogenesis of NASH[4,18,19]. Malondialdehyde (MDA), a biomarker of lipid peroxidation, is increased in murine models of NASH or patients with NASH[19,20]. While hepatic glutathione peroxidase 4 (GPX4), a key protector against lipid peroxidation, is increased in murine NASH models, drug-mediated inhibition or activation of GPX4 exacerbates or ameliorates the severity of murine NASH[19]. However, it remains unknown what cell death modes are selected in steatotic hepatocytes and the mechanism involved.

We previously found that the increase in hepatocellular death is caused by enhanced eukaryotic initiation factor 2α (eIF2α) signalling during regeneration after partial hepatectomy[8]. eIF2α signalling comprises a stress response pathway induced by various intracellular stresses that is enhanced in hepatic steatosis[21,22]. Phosphorylation of eIF2α in response to intracellular stress induces the gene expression of transcription factors such as C/EBP homologous protein (CHOP) and activating transcription factor 3 (ATF3), leading to a stress response[23]. In the present study, we investigated the mode of hepatocellular death and its regulatory mechanism during liver regeneration after liver damage by use of hepatectomised hepatic steatosis mice as models of acute liver damage and by use of MCD-induced NASH mice as models of chronic liver damage. We found that the mode of hepatocellular death switches via ATF3-dependent RIPK3 induction from apoptosis to necroptosis with exacerbation of steatosis in obese mice fed a high-fat diet (HFD) after partial fatty liver resection. We also revealed that this ATF3-dependent RIPK3 induction plays an important role in the pathogenesis not only of post-hepatectomy acute liver damage in hepatic steatosis but also of chronic liver damage in MCD-fed NASH mice and in patients with NASH.

## Results

### Hepatocellular death is increased in severe steatosis after hepatectomy

We first examined the modal change of hepatocellular death during fatty liver regeneration after acute liver damage. Specifically, in 2-week or 16-week HFD-fed obese mice, we performed 70% partial hepatectomy, which is considered a 'clean' model of acute liver

regeneration[5]. The liver histology exhibited moderate hepatic steatosis (30%–60% of hepatocytes affected) after 2 weeks of feeding and severe hepatic steatosis after 16 weeks of feeding, as the hepatic triglyceride levels increased over the course of the feeding (Supplementary Fig. 1a, b). In severe steatosis, but not in moderate steatosis, numerous broad foci of hepatocellular death appeared in the liver tissue 48 h after the partial hepatectomy (Fig. 1a). Plasma aminotransferase levels were higher with severe hepatic steatosis than with moderate steatosis (Fig. 1b, Supplementary Fig. 1c). Terminal deoxynucleotidyl transferase dUTP nick-end labelling (TUNEL) staining, which detects all types of cell death in liver tissue sections[7,24], showed a marked increase in dead cells in severely steatotic livers (Fig. 1c, d). In moderately steatotic livers, dead hepatocytes were observed solitarily and scattered within the lobules (Fig. 1c, d), described hereafter as 'solitary dead hepatocytes'. In contrast, in severely steatotic livers, dead hepatocytes were observed localised around the broad cell death foci, called 'perifocal dead hepatocytes' hereafter, in addition to the solitary cell death (Fig. 1c, d). The number of solitary dead hepatocytes in the lobules was significantly higher in severely steatotic livers than in moderately steatotic livers (Fig. 1d).

### Non-apoptotic cell death is increased in severe steatosis after hepatectomy

We then examined the type of cell death occurring during the post-hepatectomy regeneration process of steatotic livers. To determine the type of cell death, we performed double fluorescence staining of cleaved-caspase 3 (Cl-CASP3), which detects apoptosis, and TUNEL. We examined solitary dead hepatocytes rather than perifocal dead hepatocytes to eliminate the secondary effects of localised inflammation caused by broad hepatocellular death. Solitary dead hepatocytes in the lobules were mostly apoptotic with TUNEL/Cl-CASP3 double positivity in moderately steatotic livers, whereas Cl-CASP3-negative (Cl-CASP3⁻) non-apoptotic hepatocytes predominated in severely steatotic livers (Fig. 1e). The expression levels of Cl-CASP3 and cleaved-caspase 8 (Cl-CASP8) during the regeneration process were comparable between moderately and severely steatotic livers, indicating that apoptosis induction remained unchanged between the two (Fig. 1f). However, the expressions of RIPK3 and phosphorylation of RIPK3 and MLKL were increased in severely steatotic livers (Fig. 1f). In the analysis of the gene expression of necroptosis-related molecules, the expression of the *Ripk3* and tumour necrosis factor (*Tnf*) genes was significantly increased during the regeneration process of severely steatotic livers (Fig. 1g), whereas that of *Mlkl* and *Ripk1* showed no significant changes between moderately and severely steatotic livers (Supplementary Fig. 1d). We also measured the mRNA expression of ferroptosis-related genes in the liver with and without steatosis after partial hepatectomy and found that hepatic expression of *Gpx4* was not different between these two conditions and that of *Slc7a11*, another anti-ferroptotic gene, showed an insignificant increasing tendency in severe hepatic steatosis (Fig. 1g). Furthermore, hepatic levels of MDA, a biomarker of lipid peroxidation, which increased in hepatic steatosis after hepatectomy, were not different between moderately and severely steatotic livers (Fig. 1h). These results reveal that hepatocellular death shifts apoptosis to non-apoptotic cell death along with exacerbation of hepatic steatosis after hepatectomy and that this non-apoptotic hepatocellular death may be necroptosis.

### RIPK3 knockdown diminishes non-apoptotic cell death in severe steatosis after hepatectomy

To determine whether necroptosis is the dominant mode of cell death during the regeneration of severely steatotic livers, we used hepatocyte Ripk3 knockdown (R-KD) with hepatectomy. For hepatic knockdown, we used intravenous injection of short interfering RNA (siRNA) with a cationic lipid reagent. First, to examine the hepatic cellular specificity of the knockdown, we intravenously administered siGapdh.

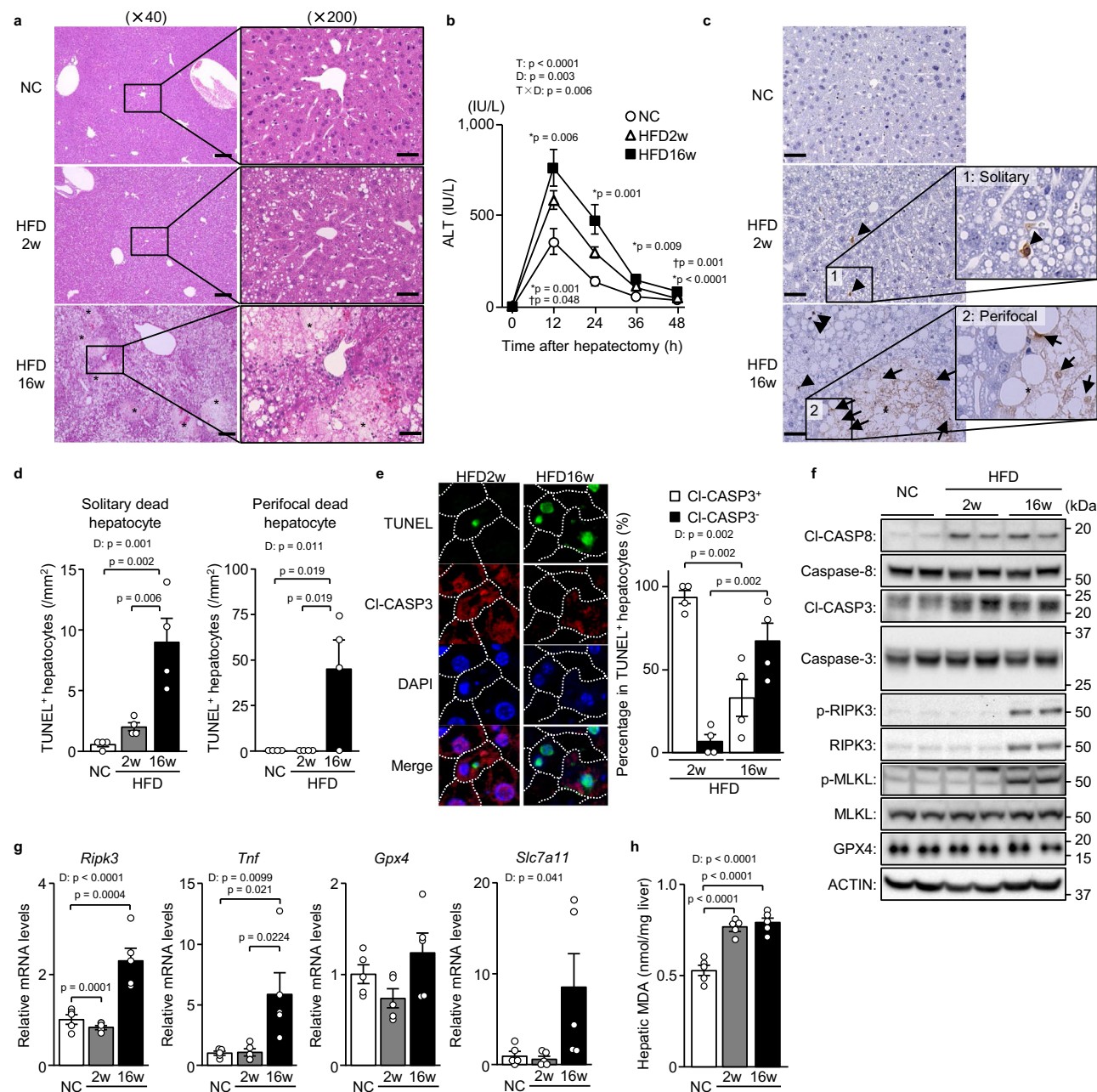

**Fig. 1 | Non-apoptotic cell death is increased in severe steatosis after hepatectomy.** Mice fed normal chow (NC) or a HFD for 2 or 16 weeks underwent partial hepatectomy. **a** Haematoxylin-eosin staining. Scale bars, 50 μm (left) and 200 μm (right). Asterisks indicate hepatocellular death foci. **b** Plasma ALT levels. * indicates p-values for HFD16w versus NC; [†] indicates *p*-values for HFD16w versus HFD2w. **c** TUNEL staining. Scale bar, 50 μm. Arrowheads indicate solitary dead hepatocytes. Asterisks indicate hepatocellular death foci. Arrows indicate perifocal dead hepatocytes. **d** Number of TUNEL[+] hepatocytes defined as described. **e** TUNEL/Cl-CASP3 double staining (left). Cl-CASP3[+] or Cl-CASP3[−] hepatocytes as a percentage of TUNEL[+] hepatocytes (right). **f** Immunoblot analysis. **g** Quantitative PCR analysis. **h** Hepatic MDA levels. Data are presented as the mean values ± SEM. [(**b**, **f–h**) $n = 5$/group; (**a** and **c–e**) $n = 4$/group, biologically independent samples]. Statistics: one-way repeated-measures ANOVA followed by Bonferroni's multiple comparisons test (**b**), one-way ANOVA followed by Tukey's multiple comparisons test (**d**, **g**, **h**), two-tailed Student's *t* test (**e**). CTRL, control; T, time effect; D, diet effect; T×D, time and diet interaction. Source data are provided as a Source Data file.

*Gapdh* mRNA was decreased in hepatocytes expressing *Alb* but not in hepatic non-parenchymal cells expressing *Adgre1*, *Cadh5* or *Des*, indicating that this knockdown displayed moderate specificity for hepatocytes in the liver (Supplementary Fig. 2a–e).

R-KD livers showed no broad hepatocellular death foci, although they were evident 48 h after hepatectomy in control livers (Fig. 2a), and lower blood aminotransferase levels compared with the control (Fig. 2b, Supplementary Fig. 2f). Moreover, there were lower levels of not only perifocal dead hepatocytes, but also solitary dead

hepatocytes scattered in the lobules (Fig. 2c, d). In R-KD livers, the ratio of Cl-CASP3[−] dead cells to the total number of solitary dead hepatocytes was drastically reduced and only Cl-CASP3-positive (Cl-CASP3[+]) apoptotic hepatocytes remained (Fig. 2e). The lower liver expression of RIPK3 did not affect the hepatic expression of Cl-CASP3 and Cl-CASP8 (Fig. 2f). However, R-KD decreased MLKL phosphorylation and *Tnf* gene expression (Fig. 2f, Supplementary Fig. 2g). These findings suggest that necroptosis is the dominant mode of hepatocellular death occurring during regeneration in severe hepatic steatosis.

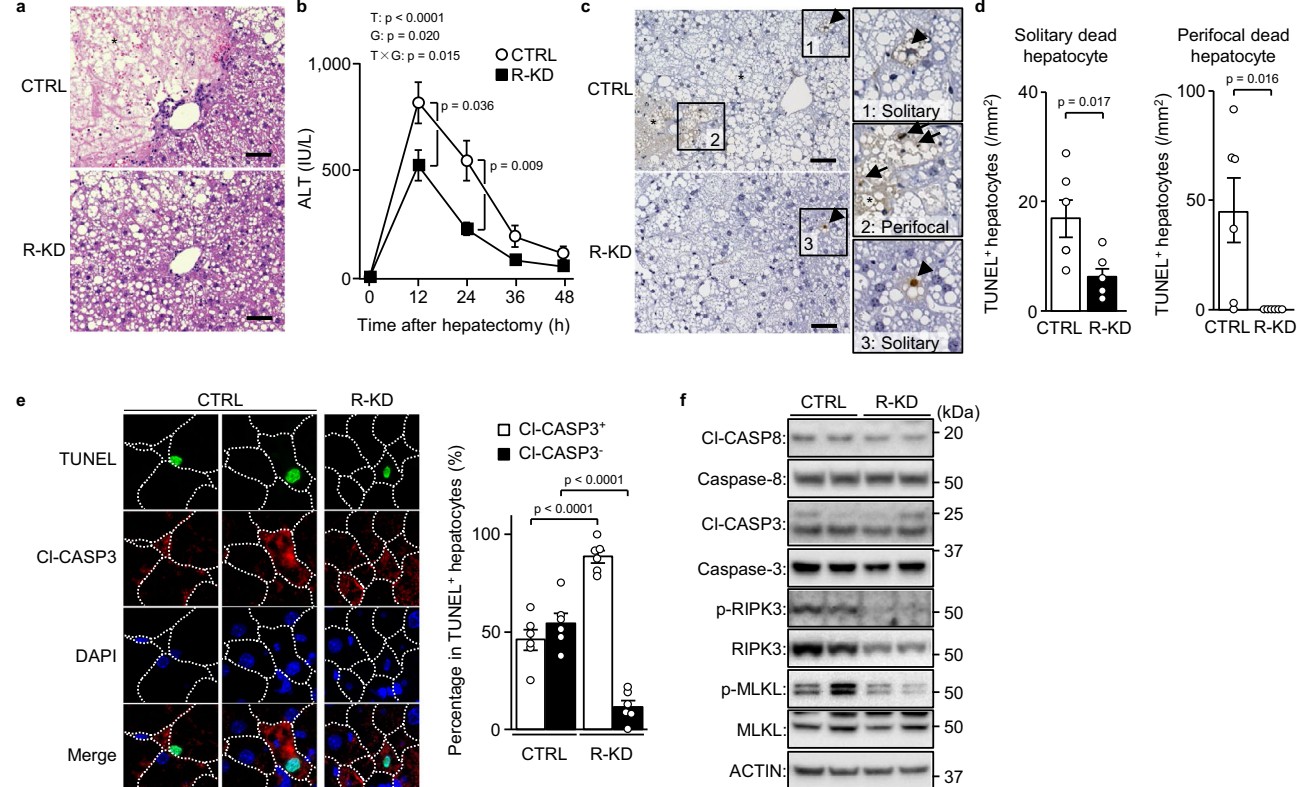

**Fig. 2 | Non-apoptotic cell death by necroptosis in severe steatosis after hepatectomy.** Mice fed a HFD for 16 weeks underwent Ripk3 knockdown (R-KD) and partial hepatectomy. **a** Haematoxylin-eosin staining. Scale bar, 50 μm. Asterisks indicate hepatocellular death foci. **b** Plasma ALT levels. **c** TUNEL staining. Scale bar, 50 μm. Arrowheads indicate solitary dead hepatocytes. Asterisks indicate hepatocellular death foci. Arrows indicate perifocal dead hepatocytes. **d** Number of TUNEL$^+$ hepatocytes defined as described. **e** TUNEL/Cl-CASP3 double staining (left). Cl-CASP3$^+$ or Cl-CASP3$^-$ hepatocytes as a percentage of TUNEL$^+$ hepatocytes (right). **f** Immunoblot analysis. Data are presented as the mean values ± SEM. [$n = 6$/group, biologically independent samples]. Statistics: one-way repeated-measures ANOVA followed by Bonferroni's multiple comparisons test (**b**), two-tailed Student's $t$ test (**d**, **e**). CTRL, control; T, time effect; G, group effect; T×G, time and group interaction. Source data are provided as a Source Data file.

## ATF3 deficiency prevents steatosis-induced necroptosis after hepatectomy

Hepatic steatosis activated eIF2α signalling even before hepatectomy (Supplementary Fig. 3a), which is triggered by intrinsic or extrinsic stresses (e.g., endoplasmic reticulum stress or hypoxia) and induces hepatocellular death during fatty liver regeneration[8]. After hepatectomy, while eIF2α phosphorylation increased CHOP expression in moderately steatotic livers, ATF3 expression was additionally elevated in severely steatotic livers (Fig. 3a, b). Accordingly, we investigated the role of ATF3 in the induction of necroptosis in severely steatotic livers after hepatectomy using hepatocyte-specific Atf3 knockout (A-KO) mice. A-KO mice showed no obvious differences in blood glucose/ insulin levels, hepatic triglyceride levels or hepatic steatosis severity versus control mice, under the pre-hepatectomy condition of 16-week HFD feeding (Supplementary Fig. 3b–e). After hepatectomy in severe hepatic steatosis, many foci of broad cell death occurred in the liver tissue of control mice, whereas no such foci were observed in A-KO mice (Fig. 3c). Moreover, A-KO mice exhibited lower blood aminotransferase levels and a marked decrease in the number of dead hepatocytes compared with control mice (Fig. 3d–f, Supplementary Fig. 3f). TUNEL/Cl-CASP3 double fluorescence staining showed that, in A-KO mice, similar to R-KD mice, Cl-CASP3$^-$ dead hepatocytes drastically decreased, and only Cl-CASP3$^+$ apoptotic hepatocytes remained (Fig. 3g). In A-KO mice, Ripk3 gene expression was significantly decreased, along with a decreasing trend in Tnf expression (Fig. 3h). Atf3 deficiency in hepatocytes did not alter eIF2α phosphorylation or CHOP/Cl-CASP3/Cl-CASP8 expression but drastically decreased RIPK3/

phosphorylated RIPK3 expression (Fig. 3i). A-KO showed unchanged expression of anti-ferroptotic gene expression and hepatic MDA levels (Fig. 3h, j).

## ATF3 overexpression increases necroptosis in un-hepatectomised severe steatosis

Next, we induced hepatocellular ATF3 overexpression in mice with severe hepatic steatosis without hepatectomy to examine the role of ATF3 in RIPK3 expression and necroptosis. We performed hepatic overexpression using intravenous injection of adenoviral vector. To examine intrahepatic cellular specificity, we used adenoviral vector encoding mCherry and found that intravenous injection of this vector enabled hepatocyte-dominant gene transfection (Supplementary Fig. 4a–e). We determined that each lobe of the liver expressed mCherry at the same level and throughout the hepatic lobe (Supplementary Fig. 4f, g).

Adenoviral vector-mediated ATF3 overexpression resulted in no difference in hepatic fat accumulation compared with the control (Supplementary Fig. 4h) but was associated with increased blood aminotransferase levels (Fig. 4a, Supplementary Fig. 4i). In severely steatotic livers, only a few dead cells could be seen in liver tissue sections unless regeneration-inducing procedures were performed, such as hepatectomy (Fig. 4b–d). Nevertheless, hepatocellular ATF3 overexpression without hepatectomy induced not only solitary hepatocyte death in the lobules, but also focal cell death, although these foci were small (Fig. 4b–d). TUNEL/Cl-CASP3 double fluorescence staining showed that most of the dead hepatocytes related to ATF3

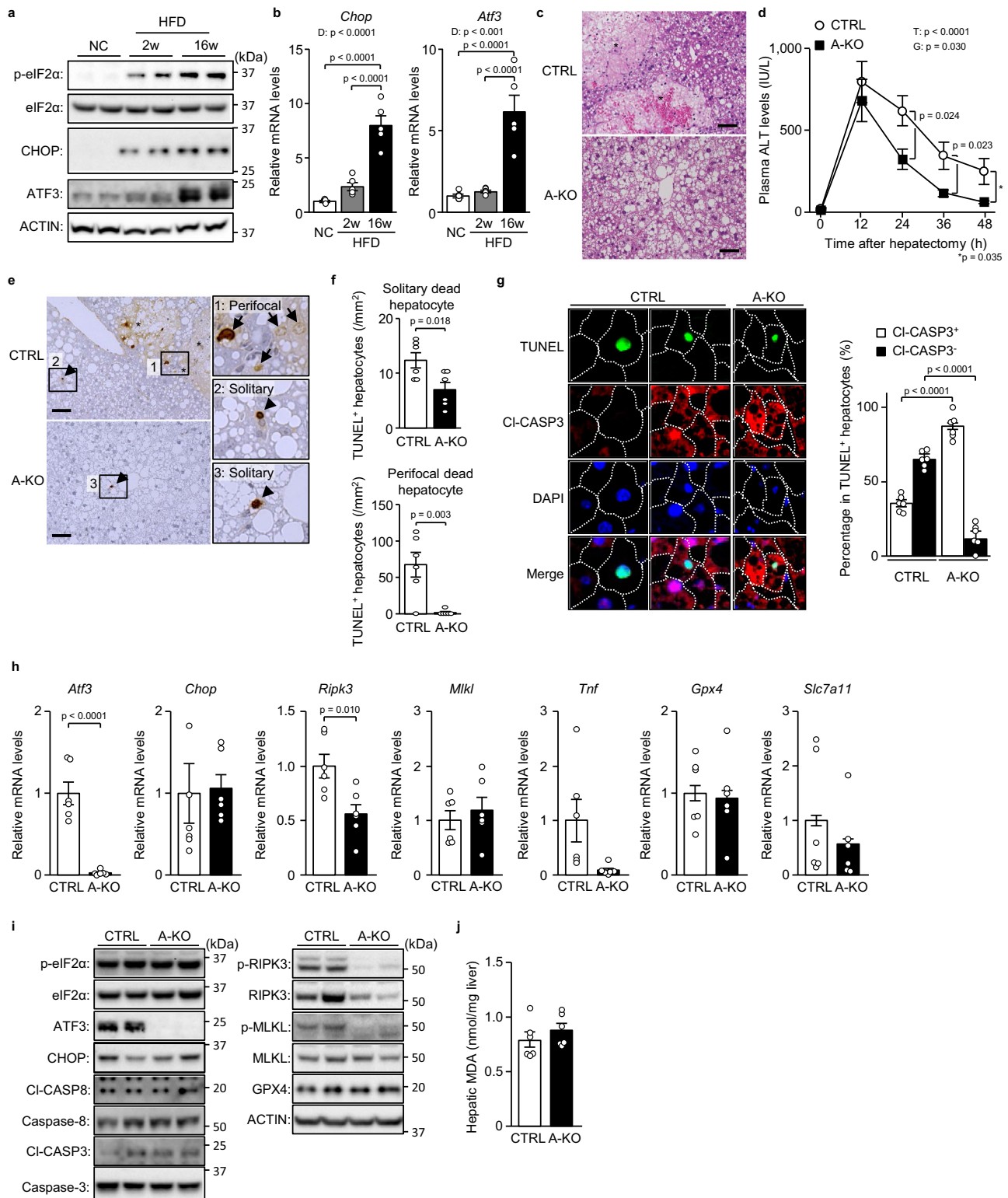

**Fig. 3 | ATF3 deficiency prevents steatosis-induced necroptosis after hepatectomy. a, b** Mice fed normal chow (NC) or a HFD for 2 or 16 weeks underwent partial hepatectomy. **a** Immunoblot analysis. **b** Quantitative PCR analysis. **c**–**j** A-KO mice and littermates (CTRL) fed a HFD for 16 weeks underwent partial hepatectomy. **c** Haematoxylin-eosin staining. Scale bar, 50 μm. Asterisks indicate hepatocellular death foci. **d** Plasma ALT levels. **e** TUNEL staining. Scale bar, 50 μm. Arrowheads indicate solitary dead hepatocytes. Asterisks indicate hepatocellular death foci. Arrows indicate perifocal dead hepatocytes. **f** Number of TUNEL$^+$ hepatocytes defined as described. **g** TUNEL/Cl-CASP3 double staining (left). Cl-

CASP3$^+$ or Cl-CASP3$^-$ hepatocytes as a percentage of TUNEL$^+$ hepatocytes (right). **h** Quantitative PCR analysis. **i** Immunoblot analysis. **j** Hepatic MDA levels. Data are presented as the mean values ± SEM. [(**a**, **b**) $n = 5$/group; (**c**–**j**) $n = 6$/group, biologically independent samples]. Statistics: one-way ANOVA followed by Tukey's multiple comparisons test (**b**), one-way repeated-measures ANOVA followed by Bonferroni's multiple comparisons test (**d**), two-tailed Student's $t$ test (**f**–**h**, **j**). D, diet effect; T, time effect; G, group effect. Source data are provided as a Source Data file.

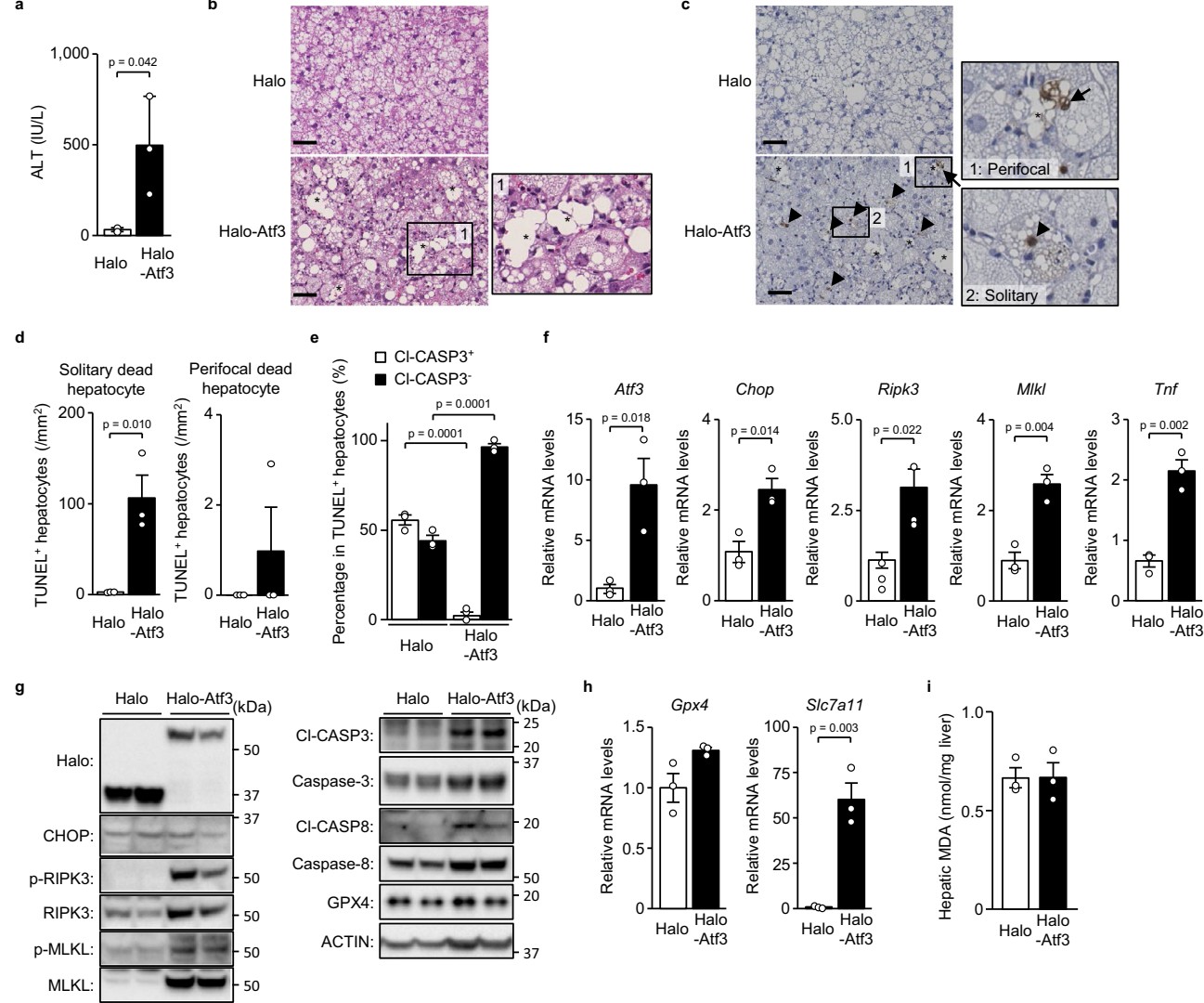

**Fig. 4 | Atf3 overexpression increases necroptosis in un-hepatectomised severe steatosis.** Halo-Atf3 or halo was overexpressed by adenovirus in mice fed a HFD for 16 weeks. **a** Plasma ALT levels. **b** Haematoxylin-eosin staining. Scale bar, 50 μm. Asterisks indicate hepatocellular death foci. **c** TUNEL staining. Scale bar, 50 μm. Arrowheads indicate solitary dead hepatocytes. Asterisks indicate hepatocellular death foci. Arrows indicate perifocal dead hepatocytes. **d** Number of TUNEL⁺ hepatocytes defined as described. **e** Cl-CASP3⁺ or Cl-CASP3⁻ hepatocytes as a percentage of TUNEL⁺ hepatocytes determined by TUNEL/Cl-CASP3 double staining. **f** Quantitative PCR analysis of genes related to the eIF2α signalling pathway and necroptosis. **g** Immunoblot analysis. **h** Quantitative PCR analysis of genes related to ferroptosis. **i** Hepatic MDA levels. Data are presented as the mean values ± SEM. [$n = 3$/group, biologically independent samples]. Statistics: two-tailed Student's $t$ test (**a**, **d**–**f**, **h**, **i**). Source data are provided as a Source Data file.

overexpression were negative for Cl-CASP3 (Fig. 4e). ATF3 overexpression resulted in increased expression of *Ripk3*, *Mlkl*, *Tnf* and *Chop* genes (Fig. 4f, Supplementary Fig. 4j), as well as RIPK3, MLKL, Cl-CASP3 and Cl-CASP8 protein (Fig. 4g). The hepatic ATF3 increase resulted in an insignificant tendency for an increase in *Gpx4* and a marked increase in *Slc7a11* (Fig. 4h). Despite the increase in these antiferroptotic genes, the hepatic ATF3 increase produced no change in the hepatic levels of MDA (Fig. 4i).

### Dephosphorylation of eIF2α decreases ATF3/RIPK3 induction after hepatectomy

To elucidate the mechanism of ATF3 induction after hepatectomy in severe hepatic steatosis, we performed hepatic overexpression of mouse GADD34, a phosphatase of eIF2α. Because we adopted gene transfection with less liver damage for the sake of hepatectomy rather than adenovirus vector-mediated transfection, we performed hepatic overexpression via intravenous injection of a chimeric adeno-associated viral vector (AAV), AAV-DJ, which enables gene transfection in the liver[8]. To examine intrahepatic cellular specificity, we used AAV encoding green fluorescent protein (GFP) and found that intravenous injection of this vector induced ubiquitous hepatocyte-dominant gene transfection throughout the liver (Supplementary Fig. 5a–g).

We performed partial hepatectomy on AAV-mediated GADD34 overexpression in mice with severe hepatic steatosis due to 16-week HFD feeding (Fig. 5a). Although GADD34 overexpression was not associated with obvious changes in blood aminotransferase levels and hepatic triglyceride levels before hepatectomy (Supplementary Fig. 5h, i), GADD34 overexpression decreased blood aminotransferase levels and broad hepatocellular death foci in the liver tissue after hepatectomy (Fig. 5b, c, Supplementary Fig. 5j). GADD34 overexpression decreased the levels of *Atf3* and *Ripk3* genes (Fig. 5d), as well as phosphorylated eIF2α, ATF3 and RIPK3 (Fig. 5a), but had no effects on MLKL protein or mRNA (Fig. 5a, d). GADD34 overexpression

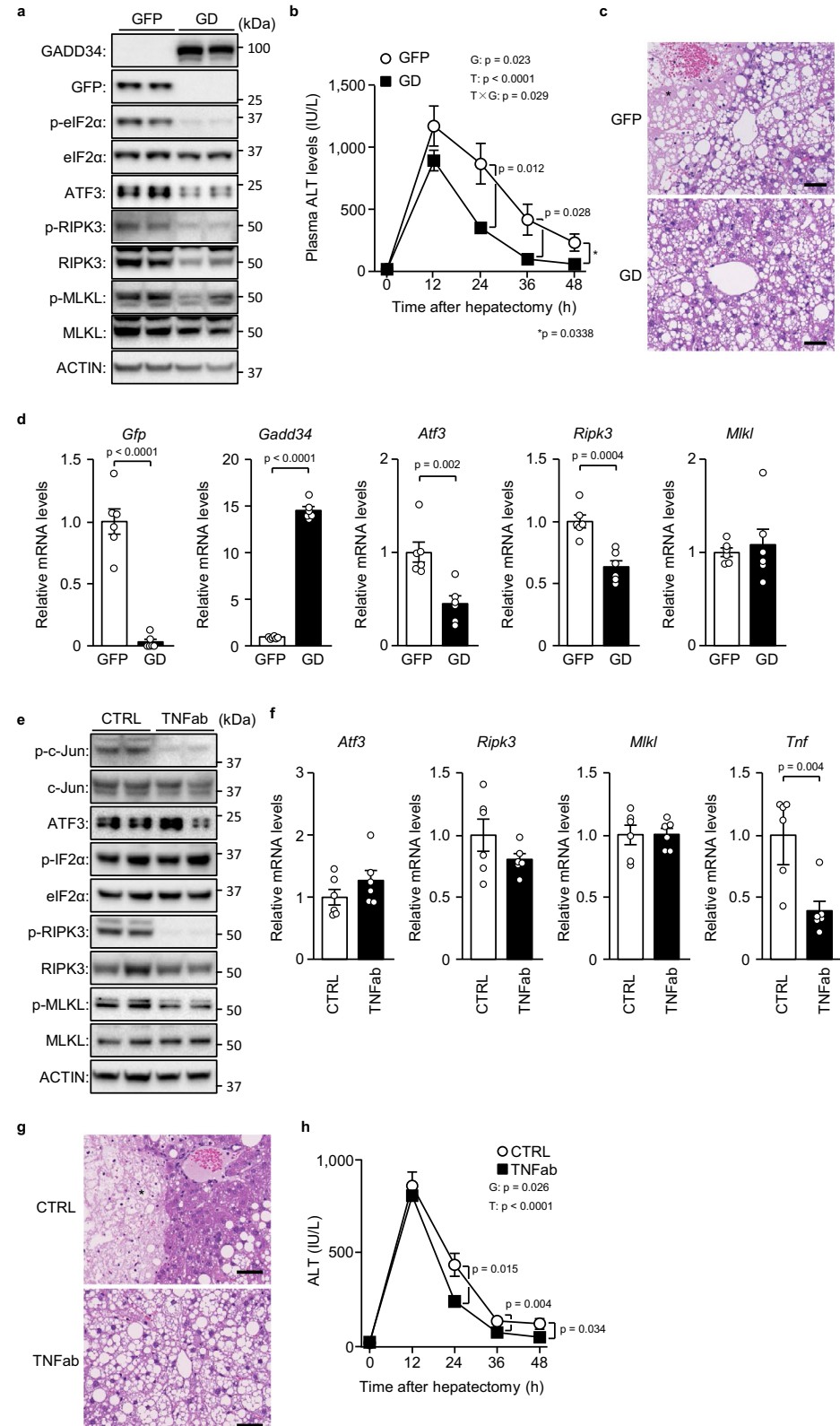

**Fig. 5 | eIF2α dephosphorylation and TNFα neutralisation in hepatic steatosis after hepatectomy. a–d** Mice with severe hepatic steatosis overexpressing GADD34 (GD) or GFP by recombinant adeno-associated virus (rAAV) underwent partial hepatectomy. **a** Immunoblot analysis. **b** Plasma ALT levels. **c** Haematoxylin-eosin staining. Scale bar, 50 μm. Asterisks indicate hepatocellular death foci. **d** Quantitative PCR analysis. **e–h** Mice with severe hepatic steatosis underwent partial hepatectomy and were injected with TNFα-neutralising antibody (TNFab) or anti-IgG (CTRL) 12 h after hepatectomy. **e** Immunoblot analysis. **f** Quantitative PCR analysis. **g** Haematoxylin-eosin staining. Scale bar, 50 μm. Asterisks indicate hepatocellular death foci. **h** Plasma ALT levels. Data are presented as the mean values ± SEM. [$n$ = 6/group, biologically independent samples]. Statistics: one-way repeated-measures ANOVA followed by Bonferroni's multiple comparisons test (**b**, **h**), two-tailed Student's $t$ test (**d**, **f**). T, time effect; G, group effect; T × G, time and group interaction. Source data are provided as a Source Data file.

resulted in no changes in anti-ferroptotic genes and the hepatic levels of MDA (Supplementary Fig. 5k, l). These results indicated that eIF2α signalling regulates ATF3-dependent RIPK3 induction in severe hepatic steatosis.

## TNFα neutralisation prevents RIPK3 phosphorylation after hepatectomy

We also examined the role of TNFα in ATF3 induction and necroptosis during severe hepatic regeneration. In particular, we intraperitoneally administered TNFα-neutralising antibody to 16-week HFD-fed mice after partial hepatectomy. TNFα neutralisation resulted in a marked decrease in the phosphorylation of RIPK3, MLKL and c-Jun (Figs. 5e, 5f) and diminished hepatic broad cell death foci after hepatectomy, along with a decrease in blood aminotransferase levels (Fig. 5g, h, Supplementary Fig. 5m). However, although TNFα neutralisation tended to slightly decrease RIPK3, it did not affect ATF3 and MLKL expression (Figs. 5e, 5f). TNFα neutralisation led to a decreased but insignificant tendency in anti-ferroptotic gene expression, but it did not affect hepatic MDA levels (Supplementary Fig. 5n, o). These results indicated that TNFα is the agent that activates RIPK3, which is induced by ATF3, after hepatectomy in severe hepatic steatosis.

## ATF3 upregulates RIPK3 transcription in hepatocytes

In isolated hepatocytes, as in mouse livers, adenoviral vector-mediated ATF3 overexpression dose-dependently increased the expressions of the *Ripk3* and *Mlkl* genes and their respective proteins (Fig. 6a, b). In isolated steatotic hepatocytes from obese/steatotic ob/ob mice, the expressions of the *Atf3*, *Ripk3* and *Mlkl* genes and their respective proteins were increased (Fig. 6c, d). Atf3 knockdown significantly reduced the expression of the *Ripk3* gene and its protein in isolated steatotic hepatocytes (Fig. 6c, d) but did not alter the expression of the *Mlkl* gene and its protein (Fig. 6c, d). Phosphorylation of MLKL and RIPK3 was not detected in these isolated hepatocytes from both lean and obese mice (Fig. 6d).

ATF3 dose-dependently increased the activity of the murine *Ripk3* promoter, which comprises the nucleotide sequence from −504 to +210 (Fig. 6e). ATF3-dependent promoter activation was induced in the sequence from −60 to +210, but not in that from −12 to +210 (Fig. 6f). The murine *Ripk3* promoter contains an Sp1 binding motif in the region from −27 to −17, to which ATF3 can bind[25] (Supplementary Fig. 6a). In mutants with one or two base substitutions of this Sp1 binding motif in the *Ripk3* promoter region from −504 to +210, ATF3-dependent promoter activation was significantly reduced (Fig. 6f, Supplementary Fig. 6a, b). Indeed, ATF3 bound to the region from −139 to −9 of the *Ripk3* promoter in a chromatin immunoprecipitation assay (ChIP) (Fig. 6g).

## ATF3 switches apoptosis to necroptosis in hepatocytes

We examined the effects of ATF3 on *Ripk3* expression using H4IIE rat hepatoma cells. In cancer cell lines, methylation of the *Ripk3* promoter region inhibits the induction of *Ripk3* gene expression, which is restored by the demethylating agent 5-aza-2′-deoxycytidine (5-AD)[26]. H4IIE cells also showed hypermethylation in the CpG sequence near the transcription start site, which was similarly reduced by 5-AD (Supplementary Fig. 7a). In H4IIE cells, the addition of 5-AD alone or ATF3 overexpression alone did not increase *Ripk3* gene expression, whereas ATF3 overexpression in the presence of 5-AD dose-dependently induced *Ripk3* gene expression (Fig. 7a). In contrast, *Mlkl* expression was increased by ATF3 overexpression even in the absence of 5-AD, and ATF3-dependent induction of *Mlkl* expression was enhanced by the addition of 5-AD (Supplementary Fig. 7b).

We then used live-cell imaging of H4IIE cells to examine whether ATF3-induced RIPK3 expression actually induces necroptosis. We used a biosensor termed SMART (a sensor for MLKL activation by RIPK3 based on fluorescence resonance energy transfer [FRET]) as a probe

for detecting necroptosis and SYTOX as a probe for detecting cell death[27]. SMART can specifically detect necroptosis as a fluorescent colour shift, while SYTOX, which is a nucleic acid stain that only passes through disrupted cell membranes, can detect cell death as fluorescence. In H4IIE cells stably expressing SMART, similar to ATF3-dependent induction of RIPK3 expression (Fig. 7a), the addition of 5-AD alone or ATF3 overexpression alone did not result in SYTOX positivity, whereas ATF3 overexpression in the presence of 5-AD resulted in SYTOX positivity and cell death induction (Fig. 7b). ATF3 overexpression in the presence of 5-AD resulted in FRET-induced colour change and cell expansion, followed by SYTOX positivity (Fig. 7c, d, Supplementary Fig. 7c, Supplementary Movie 1). While TNFα evoked cell death in H4IIE cells in the presence of 5-AD, ATF3 overexpression induced cell death from earlier time points (Fig. 7e). TNFα stimulation induced FRET⁻ cell death in H4IIE cells with cell shrinkage, a characteristic of apoptosis (Fig. 7f–h, Supplementary Fig. 7d, Supplementary Movie 2). In contrast, under ATF3 overexpression, TNFα stimulation resulted in increased cell death, particularly FRET⁺ cell death with cell expansion (Fig. 7f, i, j, Supplementary Fig. 7d, Supplementary Movie 2). With the addition of the RIPK3 inhibitor GSK872, which decreased RIPK3 and MLKL phosphorylation and increased Cl-CASP8 levels (Supplementary Fig. 7e), TNFα stimulation did not induce FRET⁺ cell death, even with ATF3 overexpression, but did induce FRET⁻ cell death with cell shrinkage (Fig. 7f, k, l, Supplementary Fig. 7d, Supplementary Movie 2). These findings indicate that the presence of ATF3 switches the mode of hepatocyte death from apoptosis to necroptosis.

## RIPK3 knockdown ameliorates NASH induced by MCD feeding

Next, we examined the role of ATF3-dependent RIPK3 induction in a model of chronic steatotic liver damage and regeneration, namely, a not hepatectomised, but diet-induced NASH model[13]. We compared eIF2α signalling activity between NASH models of chronic liver damage and hepatectomised hepatic steatosis models of acute liver damage. While both high-fat high-cholesterol diet (HFHC) feeding for 28 weeks and MCD feeding for 6 weeks result in NASH with hepatic inflammation and fibrosis[28], MCD feeding evoked a sufficiently potent activation of eIF2α signalling to induce the protein and mRNA expression of ATF3 and RIPK3, as in hepatectomised HFD-induced hepatic steatosis, but not with HFHC feeding (Fig. 8a, b). RIPK3 and MLKL phosphorylation was also increased by MCD feeding (Fig. 8a). Therefore, we investigated the involvement of ATF3 and RIPK3 in an MCD-induced NASH model, which had fatty liver, elevated blood aminotransferase levels, hepatocellular death, fibrosis-related gene expression and fibrosis (Supplementary Fig. 8a–f). In situ hybridisation (ISH) in serial sections of liver tissue detected *Ripk3* in 55.5% of *Atf3*-expressing cells and *Atf3* in 73.5% of *Ripk3*⁺ cells (Fig. 8c, d), indicating that RIPK3 induction is also closely related to ATF3 expression in the MCD-NASH model.

To examine the role of RIPK3 and necroptosis in chronic liver damage, we performed hepatocyte R-KD in MCD-fed mice. R-KD reduced blood aminotransferase levels and the number of dead hepatocytes (Fig. 8e, f, Supplementary Fig. 8g). Mice with R-KD exhibited milder fibrosis with lower expression of fibrosis-related genes, including *Tgfb1*, *Acta2* and *Col1a1* (Fig. 8g, h). While eIF2α signalling activation, including CHOP and ATF3 expression, remained unaltered by R-KD, the expression of *Tnf* fell in R-KD liver (Fig. 8i, j). As reported previously regarding systemic RIPK3 deficiency[16], hepatocyte R-KD revealed that RIPK3 and necroptosis play important roles in the chronic liver damage induced by MCD feeding.

## ATF3 knockout prevents MCD-induced NASH

To examine whether ATF3-induced RIPK3 expression is involved in chronic liver damage, we investigated the effects of MCD feeding in A-KO. A-KO mice showed no significant differences in body weight,

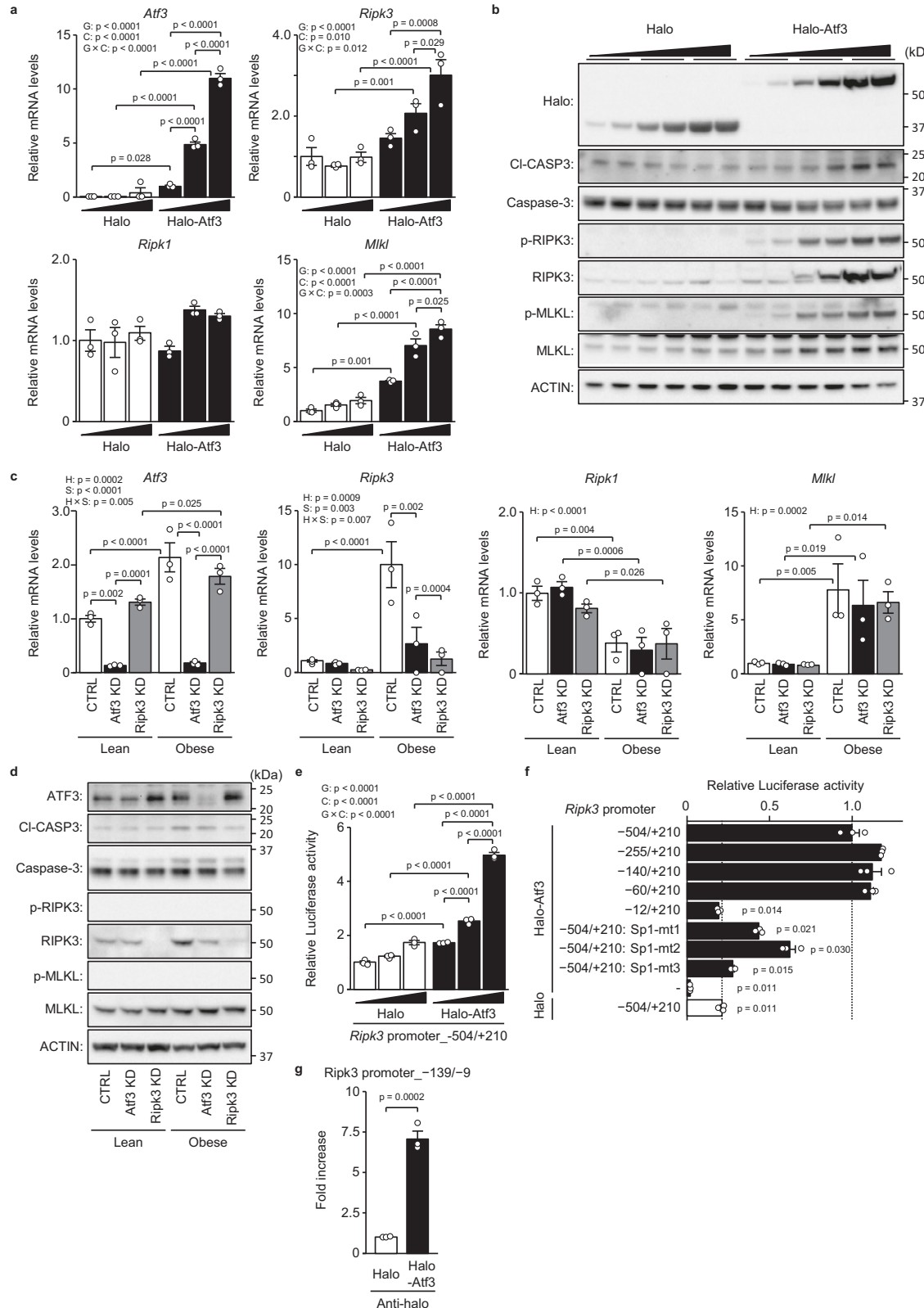

blood glucose/insulin levels or hepatic triglyceride levels compared with control mice after MCD feeding (Supplementary Fig. 9a–d). Nevertheless, A-KO mice exhibited reduced hepatic fibrosis, as assessed by collagen staining, reduced cell death, as assessed by TUNEL-stained liver histology, and lower blood aminotransferase levels (Fig. 9a–c, Supplementary Fig. 9e). In A-KO mice, in addition to decreased expression of liver fibrosis-related genes, the *Ripk3* and *Tnf*

genes, as well as RIPK3 protein expression and RIPK3 and MLKL phosphorylation, were decreased compared with control mice (Fig. 9d–f, Supplementary Fig. 9f). While MCD feeding increased both anti-ferroptotic gene expression and hepatic MDA levels, as reported previously[19] (Supplementary Fig. 9g, h), these remained unchanged by hepatic ATF3 deficiency (Fig. 9g, h). Considering the effects of A-KO, along with those of R-KD, ATF3-dependent induction of RIPK3 is

**Fig. 6 | ATF3 upregulates RIPK3 transcription in hepatocytes. a, b** Halo-Atf3 or halo was overexpressed by adenovirus in primary hepatocytes derived from lean mice. **a** Quantitative PCR analysis. **b** Immunoblot analysis. **c, d** Atf3 and/or Ripk3 were knocked down in primary hepatocytes derived from lean and obese (ob/ob) mice. **c** Quantitative PCR analysis. **d** Immunoblot analysis. **e** Analysis of the Ripk3 promoter activity in H4IIE cells by the co-transfection of luciferase reporter plasmid and three doses of halo-Atf3 or halo expression plasmid. **f** Analysis of the Ripk3 promoter activity with the indicated promoter region or Sp1 binding site-mutant by halo-Atf3 in rat H4IIE cells. **g** ChIP assay using primary hepatocytes overexpressing halo-Atf3 or halo by adenovirus. Data are presented as the mean values ± SEM. [$n$ = 3/group, biologically independent samples]. Statistics: two-way ANOVA followed by Bonferroni's multiple comparisons test (**a, c, e**), one-way ANOVA followed by Dunnett's test (**f**), two-tailed Student's $t$ test (**g**). CTRL, control; G, group effect; C, concentration effect; G × C, group and concentration interaction. H, hepatocyte group effect; S, siRNA effect; H × S, hepatocyte group and siRNA interaction. Source data are provided as a Source Data file.

closely associated with the pathogenesis of chronic liver damage induced by MCD feeding.

### Hepatic ATF3 and RIPK3 expression in patients with NASH

Finally, we explored the potential relationship of ATF3 with RIPK3 expression in the hepatocytes of human NASH by immunostaining serial sections of liver biopsy samples. The frequency of ATF3[+] hepatocytes was correlated with that of RIPK3[+] or phosphorylated RIPK3[+] hepatocytes (Fig. 10a). Furthermore, we identified the co-expression of ATF3 and RIPK3 in NASH in the same hepatocytes in serial sections of liver tissue with severe fibrosis (stage 3–4). Indeed, 74.1% of RIPK3[+] hepatocytes expressed ATF3 and 63.7% of ATF3[+] hepatocytes expressed RIPK3 (Fig. 10b, c). The frequency of phosphorylated RIPK3[+] hepatocytes was correlated with the plasma levels of AST, but not those of ATF3[+] or RIPK3[+] hepatocytes (Fig. 10d, Supplementary Fig. 10a, b). The frequencies of ATF3[+], RIPK3[+] and phosphorylated RIPK3[+] hepatocytes were not related to NASH activity, which was determined using a non-alcoholic fatty liver disease (NAFLD) activity score (NAS) ≥ 5[29] (Fig. 10e). NAS comprises the histological features of steatosis, hepatocellular damage in the form of swelling (ballooning) and lobular inflammation[30]. The frequency of either ATF3[+] or RIPK3[+] hepatocytes increased in accordance with an increase in the scores of ballooning, but not that of steatosis (Fig. 10f, g). ATF3[+] or RIPK3[+] hepatocytes tended to increase in accordance with an increase in the inflammation score, although the insignificance of this association was probably due to the low number of patients with high inflammation (Fig. 10h). The frequencies of ATF3[+], RIPK3[+] and phosphorylated RIPK3[+] hepatocytes also increased according to fibrosis stage (Fig. 10i). These results indicated that ATF3-dependent RIPK3 induction would play an important role in the hepatocellular damage in patients with NASH.

## Discussion

In hepatic steatosis, apoptosis and non-apoptotic lytic cell death, which includes necroptosis and ferroptosis, often occur side by side[1]. Because different modes of cell death have differing effects on the surrounding tissue (e.g., inflammatory responses), the predominant mode of hepatocellular death determines the condition of liver diseases[4,7,14]. In particular, lytic cell death, which promotes inflammation, acts as a trigger of acute and chronic liver damage in hepatic steatosis[4,7]. In acute liver damage, patients with apoptosis dominance have a lower mortality than patients with non-apoptosis dominance[31]. In NAFLD, the severity of the overall cell death, including lytic cell death, more closely reflects the disease condition than apoptosis alone[32]. Despite the importance of the lytic cell death of hepatocytes, it remained unclear what kind of lytic cell death is selected, and how, in steatotic hepatocytes after acute or chronic liver damage. In this study, we investigated the modes of hepatocellular death in hepatectomised mice with HFD-induced hepatic steatosis as models of acute liver damage and in MCD-induced NASH mice as models of chronic liver damage. We found that the mode of hepatocellular death shifts from apoptosis to necroptosis as the hepatic steatosis is exacerbated and that the eIF2α signalling-inducing transcription factor ATF3 determines this modal shift through RIPK3 induction. We also determined that ATF3-dependent RIPK3 induction plays important roles both in acute injury of steatotic liver and NASH.

The present study demonstrated that necroptosis plays a major role as the mode of hepatocellular death in severe hepatic steatosis. In necroptosis, cell death is executed by membrane disruption followed by MLKL phosphorylation, which is caused by RIPK3 phosphorylation. We identified the importance of necroptosis in severe hepatic steatosis through loss-of-function of hepatocellular RIPK3. Indeed, R-KD decreased MLKL phosphorylation and TUNEL[+] hepatocellular death in severe steatotic liver with acute liver damage after hepatectomy and in severe steatotic liver with chronic liver damage from MCD feeding. These finding are consistent with the phenotype of systemic RIPK3 knockout, which is reported to prevent liver damage induced by acetaminophen, alcohol and MCD feeding[15,16,33]. Although the rate of TUNEL[+] hepatocellular death was too low to be evaluated by TUNEL/Cl-CASP3 double fluorescence staining in MCD-induced NASH mice, RIPK3 knockdown markedly decreased TUNEL[+]/Cl-CASP3[−] non-apoptotic hepatocytes after hepatectomy. These results suggest that necroptosis is the predominant mode of non-apoptotic hepatocellular death during the regeneration of severe fatty liver after acute and chronic damage.

The importance of necroptosis as a cause of liver damage has not been established, despite the phenotypes of systemic RIPK3 knockout described above[15,16,33]. The unestablished role of necroptosis in the liver is attributed to the unknown mechanism underlying the induction of RIPK3 expression, which is too low in healthy liver to trigger necroptosis[4,34]. In this study, we elucidated that ATF3 is the inducer of RIPK3, but not the activator, and the determinant of the modal shift from apoptosis to necroptosis in severe hepatic steatosis. Expression analyses of serial liver sections revealed that ATF3 and RIPK3 tended to be expressed in the same hepatocytes in MCD-induced NASH mice and human patients with NASH. In vivo experiments revealed that gain- and loss-of-function of hepatic ATF3 respectively increased and decreased RIPK3 expression in mice. Furthermore, in in vitro analyses of cultured hepatocytes, ChIP demonstrated that ATF3 directly bound to the RIPK3 gene promoter region, luciferase assay analysis of RIPK3 promoter activity revealed that ATF3 regulated RIPK3 gene expression and hepatocyte live-cell imaging showed that ATF3 switched the cell death mode from apoptosis to necroptosis. Indeed, ATF3 is increased in acute liver damage induced by acetaminophen or alcohol[35,36], as is RIPK3[15,33]. Here, ATF3 overexpression increased MLKL expression, but ATF3 deficiency failed to suppress its expression, both in vivo and in vitro. ATF3 may have a limited role in the regulation of MLKL expression.

Both ATF3 and RIPK3 expression are low before hepatectomy in HFD-fed mice, even with severe hepatic steatosis induced by 16 weeks of feeding. Furthermore, 28-week HFHC feeding, which is used to create a NASH model[28], was associated with low expression of ATF3 and RIPK3. Under these conditions, the hepatocellular stress may be too weak to induce ATF3 and RIPK3. In this study, we found that hepatic ATF3 induction depends on eIF2α signalling, which is a stress response pathway induced by various intracellular stresses. Indeed, inhibition of the eIF2α signalling pathway by overexpression of GADD34, a phosphatase of eIF2α, decreased the expression of ATF3 and RIPK3 after resection of severe hepatic steatosis. In addition to eIF2α signalling, JNK-c-Jun signalling also induces ATF3 expression[37,38]. TNFα neutralisation decreased c-Jun phosphorylation, but not ATF3 expression, after hepatectomy in HFD-induced severe hepatic

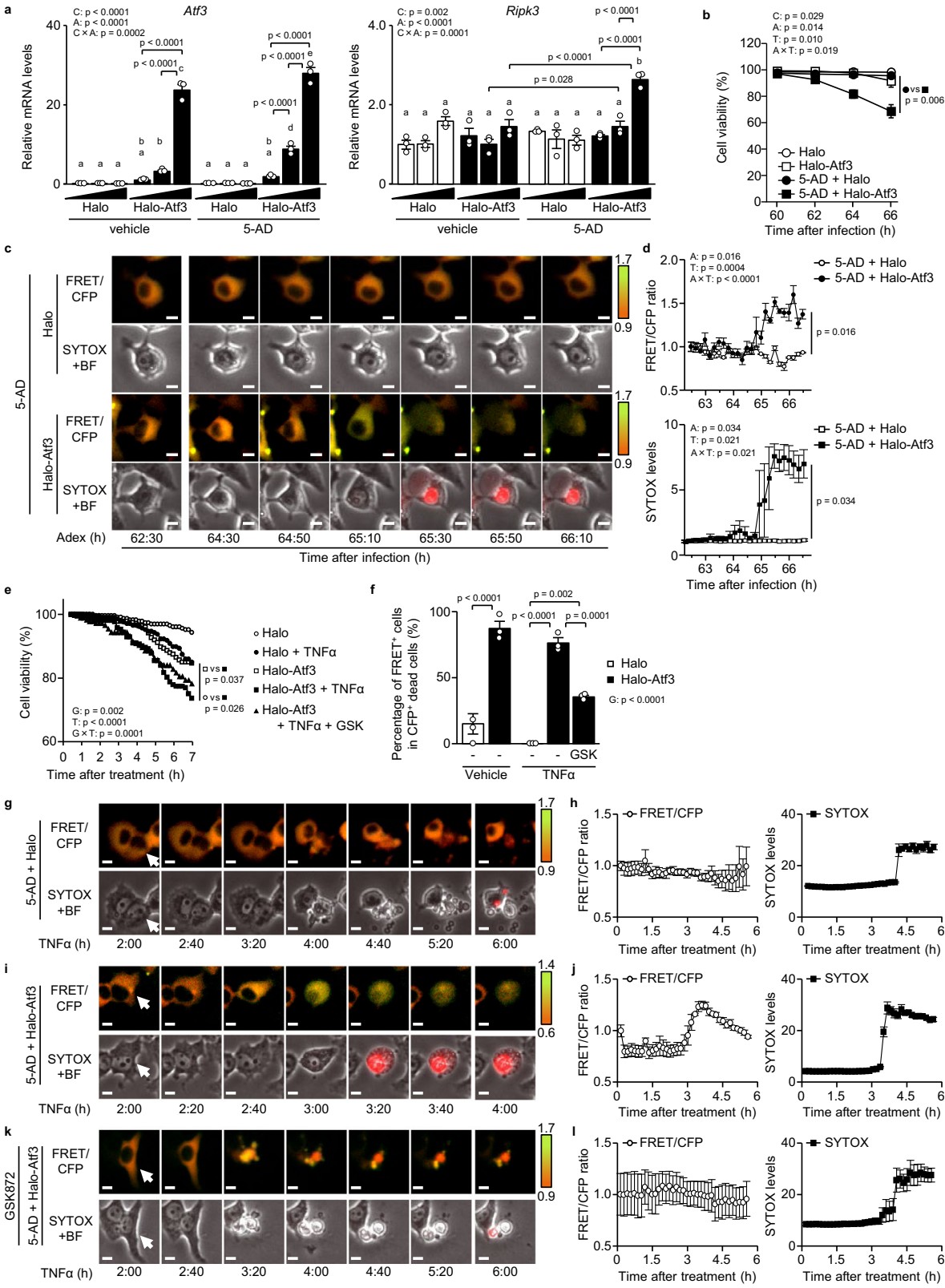

steatosis, indicating that JNK-c-Jun signalling may have an insignificant effect on ATF3 induction in this model. Previous work indicated that hepatic ATF3 knockdown decreased the levels of hepatic lipogenic genes and triglycerides in cultured hepatocytes treated with palmitate and in the liver of Zucker diabetic fatty (ZDF) rat[39,40]. Here, ATF3 deficiency produced no change in hepatic triglyceride levels in pre-hepatectomised HFD-fed mice, probably due to their milder

phenotype of obesity and hyperglycaemia compared with ZDF rat. However, given that ATF3 increases hepatic lipogenesis and lipid accumulation[39,40], ATF3 may exacerbate NASH through a synergistic effect on lipogenesis and hepatocellular death in the presence of severe obesity and hyperglycaemia.

Ferroptosis, an iron-dependent type of lytic cell death, is mediated by lipid peroxidation, and glutathione and GPX4 reduce lipid peroxides

**Fig. 7 | ATF3 switches apoptosis to necroptosis in hepatocytes. a** Halo-Atf3 or halo was overexpressed by adenovirus in H4IIE cells treated with/without 5-AD. Expression levels of *Atf3* and *Ripk3* mRNA. **b**–**d** Halo-Atf3 or halo was overexpressed by adenovirus in H4IIE-SMART cells treated with/without 5-AD. Cell death and necroptosis were monitored by SYTOX and FRET/CFP, respectively. **b** Percentage of cell viability. **c** Time-lapse images of the ratio of a single cell. Scale bar, 5 μm. **d** FRET/CFP ratio (top) and SYTOX levels (bottom) of H4IIE-SMART cells. **e**–**l** Halo-Atf3 or halo was overexpressed by adenovirus in H4IIE-SMART cells treated with 5-AD. H4IIE-SMART cells were stimulated with TNFα for 7 h and GSK872 for 24 h. Cell death and necroptosis were monitored by SYTOX and FRET/CFP, respectively. **e** Percentage of cell viability. *p < 0.05, for Halo-Atf3 versus Halo-Atf3 + TNFα.

**f** Percentage of FRET⁺ cells in CFP⁺ dead cells. **g**–**l** Time-lapse images of a single cell (**g**, **i** and **k**) and the FRET/CFP ratio (left of **h**, **j** and **l**) and SYTOX levels (right of **h**, **j** and **l**) of H4IIE-SMART cells. Data are presented as the mean values ± SEM. [*n* = 3/ group, biologically independent samples]. Statistics: two-way ANOVA followed by Tukey's multiple comparisons test (**a**), two-way repeated-measures ANOVA followed by Bonferroni's multiple comparisons test (**b**), one-way repeated-measures ANOVA followed by Tukey's multiple comparisons test (**d**, **e**), one-way ANOVA followed by Tukey's multiple comparisons test (**f**). BF, bright-field; C, chemical effect; A, adenovirus effect; C × A, chemical and adenovirus interaction; T, time effect; A × T, adenovirus and time interaction. Source data are provided as a Source Data file.

and inhibit ferroptosis[4]. ATF3 is reported to promote ferroptosis by decreasing the expression of Slc7a11, which encodes a cystine transporter and which is essential for glutathione synthesis, in a human sarcoma cell line[41]. In this study, we found that hepatectomy of steatotic liver or MCD feeding increased ferroptosis, as indicated by the elevated MDA levels and as reported previously[19]. While ferroptosis contributes somewhat to the cell death, the reduction in cell death with R-KD revealed that the dominant cell death was necroptosis under these conditions of acute and chronic hepatic steatosis damage. Furthermore, neither knockout nor overexpression of ATF3 produced any change in hepatic MDA levels in vivo, indicating that ATF3 plays only a limited role in the induction of ferroptosis in the liver of the models used in this study. A relatively short period of feeding with an MCD or choline-deficient, ethionine-supplemented diet induces ferroptosis[19,42], indicating that ferroptosis may play an important role in the early phase of NASH. In our results, ATF3 overexpression increased the expression of *Gpx4* and *Slc7a11* in addition to *Tnf*, while TNFα neutralisation decreased their expression. TNFα may induce the expression of these anti-ferroptotic genes in hepatocytes, in line with a previous report that TNFα protects against ferroptosis by inducing anti-ferroptotic genes in fibroblasts[43].

While ATF3 is the inducer of RIPK3, TNFα is a vital activator of RIPK3 in hepatic steatosis. Indeed, neutralisation of TNFα diminished the phosphorylation of RIPK3 but did not affect ATF3 expression. Furthermore, in isolated hepatocytes, ATF3 expression induced RIPK3 expression, but did not always induce phosphorylation of RIPK3 and MLKL. These results indicate that RIPK3 induction by ATF3 and its activation by TNFα are produced independently in hepatic steatosis. RIPK3 was phosphorylated only by ATF3 overexpression in isolated hepatocytes and H4IIE cells, possibly due to an unknown mechanism related to adenovirus vector-mediated transfection.

Given that TNFα leads to RIPK3 activation and necroptosis induction, which increases inflammation and TNFα levels[4,9], ATF3-dependent induction of RIPK3 in severe hepatic steatosis can cause a vicious cycle of necroptosis and inflammation. Indeed, hepatocellular deficiency of RIPK3 or ATF3 decreased hepatic TNFα expression and ATF3 overexpression increased its expression. This vicious cycle can play an important role in the pathogenesis of the exacerbation of liver damage. In conclusion, we have elucidated that ATF3-dependent induction of RIPK3 causes a modal shift of hepatocellular death from apoptosis to necroptosis and plays an important role in the regulation of hepatocellular death in acute fatty liver injury and NASH. This mechanism of RIPK3 induction and activation may be a novel therapeutic target for acute and chronic liver damage in severe hepatic steatosis.

## Methods
### Mouse experiments
All experiments involving mice were conducted in accordance with the Guide for the Care and Use of Laboratory Animals, eighth edition, and were approved by the Committee for Ethical Use of Experimental Animals of Kanazawa University, Kanazawa, Japan (approval no. AP-194033). Seven-week-old male C57BL/6J and C57BL/6JHamSlc-ob/ob

mice were obtained from Japan SLC (Shizuoka, Japan). Atf3^flox/flox mice were generated previously[44] and were crossed with albumin-Cre mice expressing Cre recombinase in hepatocytes[45] to generate liver-specific A-KO mice. Littermates were used as controls. Mice were housed with littermates in groups of two to five until the end of the experiments. The mice were maintained in a temperature- and humidity-controlled environment (22 °C ± 3 °C and 35% ± 5%, respectively) with a 12-h light/ dark cycle (lights on at 8:45 and off at 20:45) with free access to food and water. The housing conditions were closely monitored and controlled.

For partial hepatectomy experiments, 8-week-old male C57BL/6J were fed normal chow (NC) (13.6% fat; CRF-1, Oriental Yeast Co., Ltd., Kyoto, Japan) or a HFD (60% fat; D12492, Research Diets, Inc., New Brunswick, NJ) for 2 or 16 weeks and underwent 70% partial hepatectomy at 10 or 24 weeks of age. A-KO mice aged 7–9 weeks were fed a HFD for 16 weeks and underwent 70% partial hepatectomy at 23–25 weeks of age. Plasma samples were collected every 12 h from 0 to 48 h after hepatectomy. The mice were euthanized by cervical dislocation 48 h after hepatectomy and the liver was collected and then immediately frozen in liquid nitrogen or fixed in 4% paraformaldehyde/phosphate-buffered saline (PFA) (Fujifilm Wako Pure Chemical Corp., Osaka, Japan).

For NASH model experiments, C57BL/6J or A-KO mice were fed an MCD (A02082002BR, Research Diets) for 6 weeks[46], after which the livers were collected and immediately frozen in liquid nitrogen.

### Analysis of plasma and hepatic parameters
Plasma alanine aminotransferase (ALT) levels were measured using the Transaminase CII-Test-Wako kit (Fujifilm Wako Pure Chemical Corp.). Hepatic triglyceride contents were measured using the triglyceride E-Test-Wako kit (Fujifilm Wako Pure Chemical Corp.)[47]. Hepatic MDA levels were measured using the Lipid Peroxidation (MDA) Assay Kit (Colorimetric/Fluorometric) (ab118970, Abcam, Cambridge, UK).

### Histological analysis
Mouse liver tissue was fixed in 4% PFA, and sections were stained with haematoxylin-eosin or Sirius red[48]. TUNEL staining was performed using the ApopTag® Peroxidase In Situ Apoptosis Detection Kit (Millipore, Billerica, MA). The average number of TUNEL⁺ cells was measured in ten separate fields under ×20 magnification. TUNEL/Cl-CASP3 double staining was performed using a TUNEL staining kit (Click-iT Plus TUNEL Assay for In Situ Apoptosis Detection; C10617, Thermo Fisher Scientific, Waltham, MA) and Cl-CASP3 antibody (Supplementary Table 1). Cell borders were drawn by increasing the brightness of DAPI staining to the maximum.

### Immunoblotting and quantitative PCR
Total protein extraction, total RNA extraction and cDNA synthesis were performed using the following method[49]. Liver tissues and cultured cells were homogenised in ice-cold CelLytic MT Cell Lysis Reagent (Merck KGaA, Darmstadt, Germany) with protease inhibitors. Immunoblotting was performed using the antibodies listed in

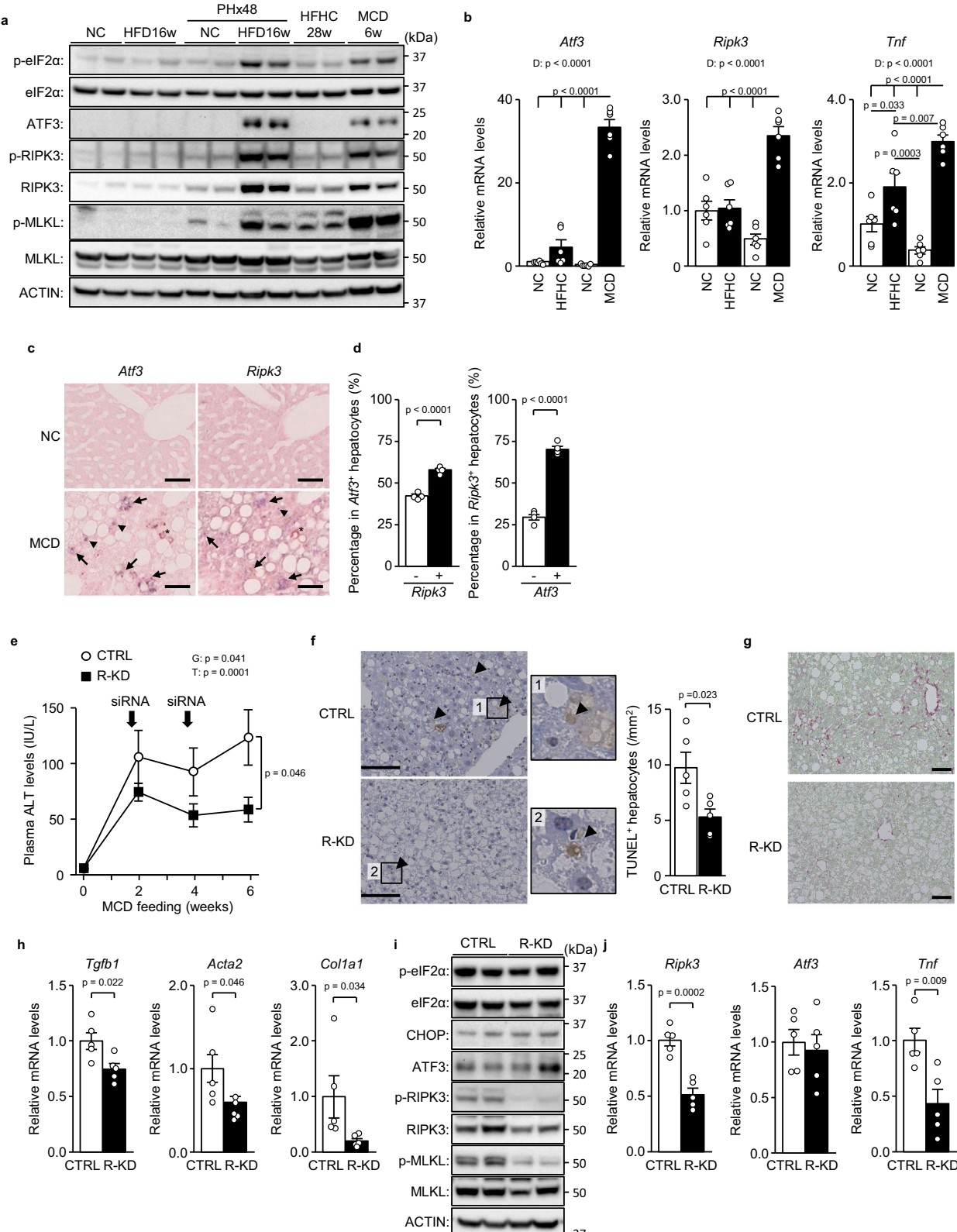

Supplementary Table 1. Protein bands were visualised using the Che-miDoc Touch Imaging System with Bio-Rad CFX Manager V1.5 (Bio-Rad, Hercules, CA). Total RNA was extracted from frozen tissue or cell samples using the SV Total RNA Isolation System (Promega, Madison, WI). cDNA was synthesised from total RNA with the PrimeScript RT reagent kit (Takara Bio, Shiga, Japan). Quantitative polymerase chain reaction (PCR) was performed using the SYBR Premix ExTaq2 kit

(Takara Bio) with *RplpO* as the control gene. The primer sets are listed in Supplementary Table 2.

### Gene knockdown with siRNA

For animal experiments, Ambion pre-designed Ripk3 siRNA or control siRNA (Thermo Fisher Scientific) was injected into the tail vein at a dose of 1 µg/g body weight using the Invivofectamine 3.0 Reagent

**Fig. 8 | RIPK3 knockdown ameliorates NASH induced by MCD feeding. a**, **b** Wild-type mice were treated as indicated. **a** Immunoblot analysis. **b** Quantitative PCR analysis. **c**, **d** ISH for *Atf3* and *Ripk3*. **c** ISH images of serial sections. Scale bar, 50 µm. Arrows indicate both Atf3⁺ and Ripk3⁺ hepatocytes. Arrowheads indicate Atf3⁺ or Ripk3⁺ hepatocytes. Asterisks indicate hemosiderin pigmentation. **d** Percentage of Ripk3⁺ or Ripk3⁻ hepatocytes in *Atf3*⁺ hepatocytes (left). Percentage of Atf3⁺ or Atf3⁻ hepatocytes in *Ripk3*⁺ hepatocytes (right). **e–j** Ripk3 siRNA was injected into the mice at 2 and 4 weeks after MCD feeding. **e** Plasma ALT levels. **f** TUNEL staining (left). Scale bar, 50 µm. Number of TUNEL⁺ hepatocytes (right). **g** Sirius red staining.

Scale bar, 50 µm. **h** Quantitative PCR analysis of genes related to fibrosis. **i** Immunoblot analysis. **j** Quantitative PCR analysis of *Ripk3*, *Atf3* and *Tnf*. Data are presented as the mean values ± SEM. [(**a**–**d**) *n* = 6/group; (**e**–**j**) *n* = 5/group, biologically independent samples]. Statistics: one-way ANOVA followed by Tukey's multiple comparisons test (**b**), two-tailed Student's *t* test (**d**, **f**, **h**, **j**), one-way repeated-measures ANOVA followed by Bonferroni's multiple comparisons test (**e**). PHx, post-hepatectomy; R-KD, Ripk3 knockdown; D, diet effect; G, group effect; T, time effect. Source data are provided as a Source Data file.

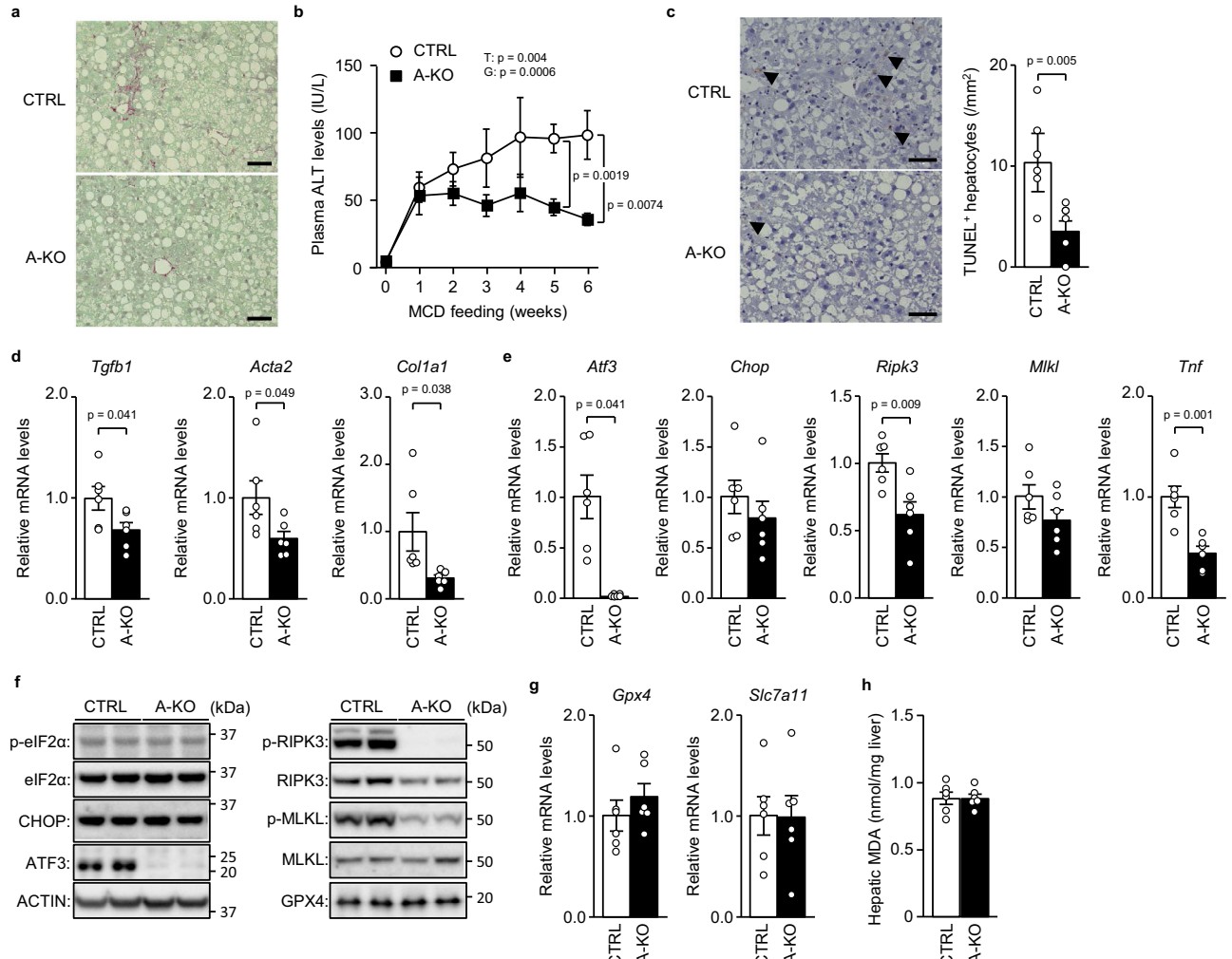

**Fig. 9 | ATF3 knockout prevents MCD-induced NASH.** A-KO mice and littermates (CTRL) were fed an MCD for 6 weeks. **a** Sirius red staining. Scale bar, 50 µm. **b** Plasma ALT levels at the indicated time points. **c** TUNEL staining (left). Scale bar, 50 µm. Arrowheads indicate TUNEL⁺ hepatocytes. Number of TUNEL⁺ hepatocytes (right). **d**, **e** Quantitative PCR analysis of genes related to fibrosis (**d**) and the eIF2α signalling pathway and necroptosis (**e**). **f** Immunoblot analysis. **g** Quantitative PCR

analysis of genes related to ferroptosis. **h** Hepatic MDA levels. Data are presented as the mean values ± SEM. [*n* = 6/group, biologically independent samples]. Statistics: one-way repeated-measures ANOVA followed by Bonferroni's multiple comparisons test (**b**), two-tailed Student's *t* test (**c**–**e**, **g**, **h**). G, group effect; T, time effect. Source data are provided as a Source Data file.

(Thermo Fisher Scientific) 24 h before hepatectomy. For cell experiments, Ambion pre-designed siRNA (Atf3, Ripk3 or control) (Thermo Fisher Scientific) was transfected into mouse primary hepatocytes using Lipofectamine RNAiMAX (Thermo Fisher Scientific).

### Adenovirus vectors
Adenovirus vectors (halo-tagged Atf3, halo-tag) were constructed by an Adenovirus Dual Expression Kit (Takara Bio)[50]. The cells and mice were used for each experiment 48–72 h and 4–5 days after adenovirus

infection, respectively. Mice fed a HFD for 16 weeks received adenoviruses (2.0 or 6.0 × 10⁸ plaque forming units) into the tail vein.

### Gene transfer with AAV vector
AAV vectors were produced using the AAV Helper-Free System (Cell Biolabs, San Diego, CA) and 293AAV cells (Agilent, Santa Clara, CA) and purified/concentrated using the ViraBind AAV Purification Kit (Cell Biolabs). Male C57BL/6J mice fed a HFD for 14 weeks were infected with AAV encoding Gadd34 or GFP as control at a dose of 2 × 10⁷ PFU/g body

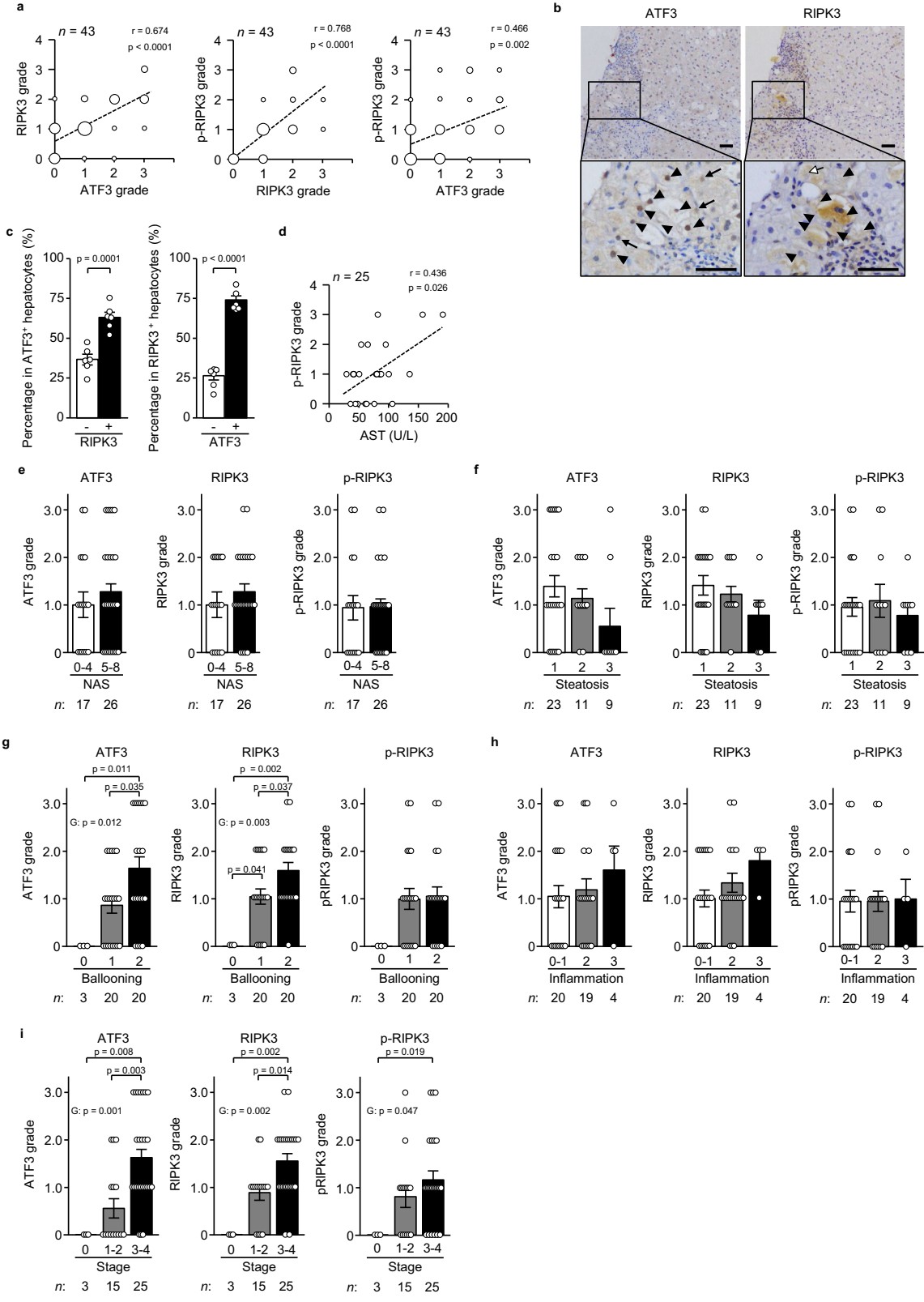

weight through the tail vein and underwent 70% partial hepatectomy 2 weeks after AAV injection.

## TNFα-neutralising antibody

TNFα-neutralising antibody (R&D Systems, Minneapolis, MN) was administered intraperitoneally at a dose of 7.0 µg/g mouse 12 h after hepatectomy, followed by liver sampling 48 h after hepatectomy.

## Cell culture

Primary hepatocytes were isolated from 8–12-week-old male C57BL/6J or C57BL/6JHamSlc-ob/ob mice and cultured using the following method[51]. The liver of anaesthetised mice was perfused at a rate of 4.5 mL/min for the first 3 min with Hank's balanced salt solution (HBSS) (Fujifilm Wako Pure Chemical Corp.) containing 10 mM HEPES-NaOH (pH 7.4) and then for 15 min with HBSS

**Fig. 10 | Hepatic ATF3 and RIPK3 expression in patients with NASH.** Serial sections of liver specimens in NAFLD patients were stained for ATF3, RIPK3 or phosphorylated RIPK3 (p-RIPK3). **a** Correlation (*r*) between the ATF3 and RIPK3 grades, between the RIPK3 and p-RIPK3 grades, and between the ATF3 and p-RIPK3 grades, calculated using Spearman's rank correlation test. **b** Images of serial sections from human NASH (stages 3 and 4). Arrowheads indicate both ATF3+ and RIPK3+ hepatocytes. Arrows indicate ATF3+ or RIPK3+ hepatocytes. Scale bar, 50 μm. **c** Percentage of RIPK3+ or RIPK3− hepatocytes in ATF3+ hepatocytes (left).

Percentage of ATF3+ or ATF3− hepatocytes in RIPK3+ hepatocytes (right). **d** Correlation (*r*) between the p-RIPK3 grade and AST level, calculated using Spearman's rank correlation test. **e–i** Semiquantitative evaluation of ATF3, RIPK3 and p-RIPK3 by NAS (**e**), steatosis (**f**), ballooning (**g**), inflammation (**h**) and fibrosis stage (**i**). Group comparisons were tested by the Kruskal−Wallis test. Data are presented as the mean values ± SEM. [(**a**, **d**–**i**) Sample size is indicated in each figure. **b**, **c** *n* = 6/group, biologically independent samples]. The liver biopsy samples are biologically independent samples. Source data are provided as a Source Data file.

containing collagenase type I (0.3 mg/mL) (Worthington, Lakewood, NY) and Protease Inhibitor Cocktail Complete-EDTA free (one tablet per 50 mL) (Roche, Basel, Switzerland). Hepatocytes from C57BL/6J were purified by density gradient centrifugation with Percoll (Merck KGaA). Isolated hepatocytes were seeded at 5.0 × 10⁴ cells/cm² and incubated in DMEM (Fujifilm Wako Pure Chemical Corp.) supplemented with 10% foetal bovine serum (FBS) (Thermo Fisher Scientifics). Hepatocytes were either infected with adenovirus or transfected with siRNA. H4IIE cells (rat hepatoma cells; ATCC, Manassas, VA) were cultured in DMEM with 10% FBS with/without 3 μM 5-Aza-2′-deoxycytidine (5-AD) (A3656, Merck KGaA) and infected with adenovirus. H4IIE cells were stimulated with 25 ng/mL TNFα (315-01 A, PeproTech, Cranbury, NJ) for 7 h and 5 μM GSK872 (S8465, Selleck, Houston, TX) for 24 h. H4IIE-SMART cells were generated using the following method[27]. H4IIE cells were transfected with pT2KXIG-mSMART (encoding mSMART) and pCAGGS-T2TP (encoding transposase) by Lipofectamine 3000 (Thermo Fisher Scientific). After transfection, ECFP+ cells were sorted by a FACSAria™ Fusion Cell Sorter (BD Biosciences, Franklin Lakes, NJ).

**Plasmids and site-directed mutagenesis**
The cDNA for halo-tagged *Atf3* was cloned by PCR and inserted into pHTN Halotag CMV-neo Vector (Promega). Reporter plasmids of various lengths and containing *Ripk3* promoter were generated using the primer sets listed in Supplementary Table 3 and inserted into pGL4.27 Vector (Promega). The point mutation in the *Ripk3* promoter was introduced into the pGL4.27-*Ripk3* promoter (−504/+210). Mutant constructs were generated using the QuickChange Site-Directed Mutagenesis kit (Agilent) with the primer sets listed in Supplementary Table 3. Construct sequences were confirmed by DNA sequencing.

**Luciferase assay**
H4IIE cells were transfected with the reporter plasmid, the expression plasmid (halo-tagged Atf3 or halo-tag) and the Renilla luciferase expression plasmid (pRL-SV40, Promega) as internal control using Lipofectamine 3000 (Thermo Fisher Scientific). Twenty-four hours after transfection, the cells were harvested and luciferase activity was measured using the Dual-Luciferase® Reporter Assay System (Promega)[52].

**Chromatin immunoprecipitation**
ChIP assays were performed using the following method[53]. Cells were crosslinked in Dulbecco's modified Eagle medium containing 1% formaldehyde (Thermo Fisher Scientific) for 10 min at 37 °C. Then, the cells were lysed after cross-linking by Dounce homogenisation (Nippon Gene, Tokyo, Japan). Nuclei were then isolated and sonicated to create chromatin fragments of approximately 200−600 bp using the Bioruptor UCD-250 (BM Bio, Tokyo, Japan). The chromatin was next incubated with HaloLink Resin (Promega) for 4 h at room temperature. To test for enrichment of Halo-ATF3-bound sites, PCR amplification of the *Ripk3* promoter was performed on a ChIP sample with/without HaloCHIP Blocking Ligand (Promega).

**Time-lapse imaging**
Time-lapse imaging was performed with a fluorescence microscope (BZ-x710, Keyence, Osaka, Japan) and a time lapse module (BZ-H3XT, Keyence), using the following method[54]. Cells were treated with 1 μM SYTOX Orange Nucleic Acid Stain (S11368, Thermo Fisher Scientific) at 37 °C in a humidified atmosphere of 5% CO₂ in air. SYTOX, CFP and CFP-YFP FRET images were captured through a TxRed filter set (Keyence, OP-87765), a CFP filter set (49001-UF1-BLA, MSQUARE, Fukuoka, Japan) and a CFP-YFP FRET filter set (49052-UF1-BLA, MSQUARE), respectively.

**In situ hybridisation**
After perfusion, mouse liver was dissected, fixed with G-Fix (Genostaff, Tokyo, Japan), embedded in paraffin and sectioned at 6 μm. ISH was performed on serial sections with an ISH Reagent Kit (Genostaff) according to the manufacturer's instructions and as previously reported[55]. Hybridisation was performed with probes (250 ng/mL) for *Ripk3* and *Atf3* in G-Hybo-L (Genostaff) for 16 h at 60 °C, and the colocalisation of *Atf3* and *Ripk3* was determined in serial sections as *Ripk3*+/− cells as a percentage of *Atf3*+ cells and *Atf3*+/− cells as a percentage of *Ripk3*+ cells. The sequences of the probes used are listed in Supplementary Table 4.

**Human patients**
Liver tissue samples were obtained from 43 patients (21 male and 22 female) diagnosed with NAFLD through ultrasound-guided percutaneous liver biopsy. All tissue specimens were collected from the hepatobiliary file of the Department of Human Pathology, Graduate School of Medical Sciences, Kanazawa University. Baseline characteristics are presented in Supplementary Table 5. Three evaluation items−steatosis (0−3), lobular inflammation (0−3) and hepatocellular ballooning (0−2)−were scored to determine the NAS by the NASH Clinical Research Network scoring system[29,30], along with fibrosis scored by the Brunt classification[56]. All patients gave written informed consent to participate in the study in accordance with the Helsinki Declaration. This study was approved by the regional ethics committee (Medical Ethics Committee of Kanazawa University, No. 2016-072(305)).

**Semiquantitative immunohistochemistry**
Human liver biopsy specimens were fixed, embedded, sectioned and stained as described below[57]. Serial sections were immunostained with ATF3, RIPK3 or phosphorylated RIPK3 antibodies (Supplementary Table 1). The Envision-HRP system (Dako Japan Co., Tokyo, Japan) was used for ATF3 and RIPK3, whereas the Novolink Polymer Detection system (Leica Biosystems, Tokyo, Japan) was used for phosphorylated RIPK3.

The expression levels of ATF3, RIPK3 and phosphorylated RIPK3 were semiquantitatively scored twice in immunostained liver sections: 0, no positive cells; 1, minimal, with a few positive cells in several regions; 2, several positive cells in several regions; and 3, considerable amount of positive cells in more than half of the regions. The ATF3, RIPK3 and phosphorylated RIPK3 grade was evaluated as the score of each component of NAS and the stage. The colocalisation of ATF3 and

RIPK3 was determined in serial sections as RIPK3$^{+/-}$ cells as a percentage of ATF3$^+$ cells and ATF3$^{+/-}$ cells as a percentage of RIPK3$^+$ cells.

## Statistics and reproducibility

Data are presented as the mean values ± SEM. Statistical analysis was performed using IBM SPSS Statistics 24 (IBM Japan, Tokyo, Japan). Group comparisons in time-course data were examined using one-way repeated-measures analysis of variance (ANOVA) followed by Bonferroni's multiple comparisons test. Group comparisons were tested by one-way and two-way ANOVA followed by Bonferroni's multiple comparisons test, Tukey–Kramer's honestly significant difference test or Dunnett's test. Student's $t$ test was performed to determine the significance between two independent groups. For semiquantitative immunohistochemical evaluations, group comparisons were tested by the Kruskal–Wallis test. Correlation coefficients were evaluated using Spearman's rank correlation test. Differences were considered statistically significant at $p < 0.05$. Animal experiments were repeated at least twice. In vitro experiments were performed in triplicate and repeated at least three times. In all experiments, attempts to replicate the experiments were successful.

## Reporting summary

Further information on research design is available in the Nature Portfolio Reporting Summary linked to this article.

## Data availability

The data that support the findings of this study are available within the article and supplementary information. Source data are provided with this paper.

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

## Acknowledgements

The authors thank C. Asahi of Kanazawa University for providing technical assistance and ThinkSCIENCE (Tokyo, Japan) for help in preparing the manuscript. This work was supported by KAKENHI Grants (JP20H04943, JP20H04102, and JP22K19545 to H.I. and JP17H06300, JP19K09192, and JP22H03507 to Y.I.) from the Japan Society for the Promotion of Science (JSPS); a Japan Diabetes Foundation grant; a Boehringer and Lilly research grant from the Japan Diabetes Foundation; the MSD Life Science Foundation, Public Interest Incorporated Foundation; the NOVARTIS Foundation (Japan) for the Promotion of Science; the Institute of Medical Science of Asahi Life Foundation; the Uehara Memorial Foundation; the Takeda Science Foundation; the Japan Foundation for Applied Enzymology; the Fuji Foundation; the Hokuriku Bank Research Grant for Young Scientists; the Kanae Foundation; the Mitsubishi Foundation; Taiju Life Social Welfare Foundation; the Naito Foundation; the Japan Agency for Medical Research and Development (AMED) through AMED-CREST (JP20gm1210002 to H.N.) and the Practical Research Project for Life-Style related Diseases including Cardiovascular Diseases and Diabetes Mellitus (JP21ek0210156 to Y.I.); and the Japan Science and Technology Agency (JST), CREST (JPMJCR2123 to Y.I.).

## Author contributions

Y.I. obtained the data, contributed to the discussion and wrote the manuscript. K.T., S.M., Y.Y., S.A., K.R., S.H. and K.H. obtained the data and contributed to the discussion. E.H., H.W., K.K. and Y.O. obtained the data. C.T., M.M., S.Ki., M.H., S.Ka., M.K. and H.N. contributed to the discussion and reviewed and edited the manuscript. H.I. researched the data, designed the study, contributed to the discussion, wrote the manuscript and is the guarantor.

## Competing interests

The authors declare no competing interests.

## Additional information

[1]Metabolism and Nutrition Research Unit, Institute for Frontier Science Initiative, Kanazawa University, Kanazawa, Japan. [2]Department of Physiology and Metabolism, Graduate School of Medical Sciences, Kanazawa University, Kanazawa, Japan. [3]Department of Biochemistry and Molecular Vascular Biology, Graduate School of Medical Sciences, Kanazawa University, Kanazawa, Japan. [4]Division of Immunology and Molecular Biology, Cancer Research Institute, Kanazawa University, Kanazawa, Japan. [5]Department of Biochemistry, Toho University School of Medicine, Tokyo, Japan. [6]Division of Oncology and Molecular Biology, Cancer Research Institute, Kanazawa University, Kanazawa, Japan. [7]Department of Molecular Metabolic Regulation, Diabetes Research Center, Research Institute, National Center for Global Health and Medicine, Tokyo, Japan. [8]Medical Research Institute, Tokyo Medical and Dental University, Tokyo, Japan. [9]Department of Gastroenterology, Graduate School of Medicine, Kanazawa University, Kanazawa, Japan. [10]Department of Clinical Laboratory Medicine, Graduate School of Medical Science, Kanazawa University, Kanazawa, Japan. [11]Division of Diabetes and Endocrinology, Department of Internal Medicine, Kobe University Graduate School of Medicine, Kobe, Japan. [12]Department of Biochemistry and Molecular Biology, University of Southern Denmark, Odense M, Denmark. [13]Center for Functional Genomics and Tissue Plasticity (ATLAS), University of Southern Denmark, Odense M, Denmark. [14]Research Center for Experimental Modeling of Human Disease, Kanazawa University, Kanazawa, Japan. [15]The Institute of Medical Science, Asahi Life Foundation, Tokyo, Japan. [16]Departments of Human Pathology, Graduate School of Medical Sciences, Kanazawa University, Kanazawa, Japan.
✉e-mail: inoue-h@staff.kanazawa-u.ac.jp

