## [Peer Review File · Nature Communications]

The transcription factor ATF3 switches cell death from apoptosis to necroptosis in hepatic steatosis in male miceREVIEWER COMMENTS

Reviewer #1 (Remarks to the Author):

In the manuscript 'ATF3 switches cell death from apoptosis to necroptosis in hepatic steatosis' the authors analyze the mechanisms that drive necroptosis during the progression of NASH. They initially show that mice with severe steatosis have increased necroptosis following hepatectomy, and this could be limited by knockdown of RIPK3, a key driver of necroptosis. The key finding of this paper is the novel regulation of necroptosis through ATF3-mediated upregulation of RIPK3, shown through mouse models and in vitro data using primary hepatocytes and an interesting model of hepatoma cells where the Ripk3 gene requires demethylation for ATF3 to promote its expression. Increased ATF3 expression during the progression of steatosis is suggested to be a switch that drives necroptosis as opposed to apoptosis. While different mouse models are utilized, and the in vitro data is thorough, several issues limit the clarity and impact of this work. The use of a hepatectomy model without clear justification distracted from the major conclusions regarding the increased necroptosis in hepatic steatosis. Indeed, necroptosis in the severe steatosis model in Figures 1 and 2 is not analyzed. In addition, key markers of necroptosis (p-MLKL) were not analyzed in several experiments, relying only on mRNA analyses of total MLKL. The potential role of ferroptosis, which is also regulated in part by ATF3, is also not investigated. The impact of the manuscript in its current form is judged to be moderate, but to a potentially limited audience.

Comments.

1. Other groups previously showed that loss of RIPK3 was protective against fibrosis in murine models of NASH (Gautheron et al 2014), and that RIPK3 and MLKL were increased in patients with NASH (Afonso and Rodrigues et al 2015). Similar to this manuscript, Afonso and Rodrigues et al utilized a 18 week HFD and observed increased necroptosis. The authors need to analyze the prevalence of necroptosis prior to partial hepatectomy to confirm their MCD mouse model.
2. While the authors here use a model where a 70% hepatectomy is performed following development of steatosis, the pathological relevance, or the impact of this model to the scientific field, is not made clear, particularly as the rest of the manuscript focuses on steatosis and NASH-driven fibrosis. Furthermore, the hepatectomy model is not repeated in other murine models of steatosis included in the paper. The lack of consistency limits the impact of the conclusions.
3. The authors analyzed MLKL in whole liver using qPCR, but did not show levels of total or phosphorylated MLKL. Indeed, p-MLKL is missing from several in vivo and in vitro analyses.
4. Both TNF and Caspase-8 are a key regulator of necroptosis in mice. The authors should 1) analyze whether the increased necroptosis is dependent on, or independent of, caspase-8 levels, and 2) determine whether ATF3 upregulation is downstream of TNF.
5. The authors should assess the potential contribution of ferroptosis in their models, particularly since ATF3 can also promote ferroptosis.
6. The use of an siRNA to reduce RIPK3 in mice, as well as the adenoviral overexpression, raises questions regarding the specificity of the results. RIPK3 has been shown to be highly expressed by cholangiocytes, as well as non-parenchymal cells in the liver, whereas overexpression could impact several hepatic cell types. This is additionally called into question, as TNF expression is reduced in MCD mice receiving siRNA targeting RIPK3, which is a canonical inducer of necroptosis.

7. In the overexpression models, a single adenoviral infection was performed. Did the expression of ATF3 persist? Was the expression preferentially in hepatocytes, or was it ubiquitous throughout the liver?
8. Recent manuscripts show that ATF3 promotes NASH (Tu et al, Hepatology 2020)– the authors need to reconcile the literature with their findings that ATF3 loss did not seem to impact steatosis development in Extended Data Figure 2.
9. The results section would greatly benefit from additional introduction of the methods/models used, particularly in vivo, in order to improve the ease of understanding and clarity for readers.

Reviewer #2 (Remarks to the Author):

The manuscript from Inaba et al. titled 'ATF3 switches cell death from apoptosis to necroptosis in hepatic steatosis' demonstrated that increased hepatic steatosis severity shifts the hepatocellular death mode from apoptosis to necroptosis, aggravating the liver damage. In vivo, ATF3 regulates the expression of RIPK3. Under severe hepatic steatosis conditions, hepatic ATF3-deficient or -overexpressing mice display decreased or increased necroptosis, respectively. In vitro studies implicate that ATF3 increases RIPK3 expression by directly binding to its gene promoter leading to necroptosis. In mice and patients with NASH disease severity appears to associate with ATF3 and RIPK3 expression in hepatocytes.

This study points out a novel function of ATF3 related to necroptosis in liver steatosis injury.

However, some issues remain to be considered:

1. The authors indicated that ATF3 is induced in hepatocytes during liver steatosis injury. What is exactly the mechanism associated with this increased expression?
2. What is the potential involvement of specific pathways governing tissue inflammation on the role of ATF3 in liver steatosis injury? Does RNA expression data suggest that inflammatory responses are one of the key pathways involved?
3. In human studies, what are the mRNA levels of ATF3 and RIPK3? The human study will need to be further strengthened by additional functional analysis between ATF3/RIPK3 levels and liver injury outcomes (e.g. ALT, AST). Moreover, what are the levels of RIPK3 phosphorylation in low and high-NAS score patients?
4. In the analysis of the gene expression of necroptosis-related molecules, expression of the *MIK1* and *Ripk1* genes was not significantly increased during the regeneration process of severely steatotic livers. Would this contradict the increased levels of necroptosis, which is triggered by the sequential

phosphorylation of RIPK3 and MLKL that disrupts the cell membrane? What additional mechanism governs RIPK3-driven necroptosis?

RESPONSES TO REVIEWER COMMENTS

Manuscript: NCOMMS-17-09259, 'ATF3 switches cell death from apoptosis to necroptosis in hepatic steatosis'

REVIEWER COMMENTS

Reviewer #1 (Remarks to the Author):

In the manuscript 'ATF3 switches cell death from apoptosis to necroptosis in hepatic steatosis' the authors analyze the mechanisms that drive necroptosis during the progression of NASH. They initially show that mice with severe steatosis have increased necroptosis following hepatectomy, and this could be limited by knockdown of RIPK3, a key driver of necroptosis. The key finding of this paper is the novel regulation of necroptosis through ATF3-mediated upregulation of RIPK3, shown through mouse models and in vitro data using primary hepatocytes and an interesting model of hepatoma cells where the Ripk3 gene requires demethylation for ATF3 to promote its expression. Increased ATF3 expression during the progression of steatosis is suggested to be a switch that drives necroptosis as opposed to apoptosis.

While different mouse models are utilized, and the in vitro data is thorough, several issues limit the clarity and impact of this work. The use of a hepatectomy model without clear justification distracted from the major conclusions regarding the increased necroptosis in hepatic steatosis.

We apologise for the lack of a thorough explanation and clear description of why we used a hepatectomised model of high-fat diet (HFD) feeding, in addition to a NASH model of methionine/choline-deficient diet (MCD) feeding. The aim of this study was to clarify the regulation of hepatocellular death during regeneration after acute and chronic liver damage in hepatic steatosis, as hepatocellular death during regeneration exacerbates both types of fatty liver damage. Because hepatectomy models are a 'clean' model for studying acute regeneration without any direct effects from the causes of liver damage (e.g., acetaminophen, carbon tetrachloride, ischaemia-reperfusion), as reported previously^{1,2}, we used a hepatectomy model to investigate hepatocellular death during hepatic regeneration after acute damage. To the Introduction, we have added the following: 'we investigated the mode of hepatocellular death and its regulatory mechanism during liver regeneration after liver damage by use of hepatectomised hepatic steatosis mice as models of acute liver damage and by use of MCD-induced NASH mice as models of chronic liver damage.' To the Results, we have added the following: 'Specifically, in 2-week or 16-week HFD-fed obese mice, we performed 70% partial hepatectomy, which is considered a 'clean' model of acute liver

regeneration⁵.' Furthermore, to the first paragraph of the Discussion, we have added the following: 'Despite the importance of the lytic cell death of hepatocytes, it remained unclear what kind of lytic cell death is selected, and how, in steatotic hepatocytes after acute or chronic liver damage. In this study, we investigated the modes of hepatocellular death in hepatectomised mice with HFD-induced hepatic steatosis as models of acute liver damage and in MCD-induced NASH mice as models of chronic liver damage. We found that the mode of hepatocellular death shifts from apoptosis to necroptosis as the hepatic steatosis is exacerbated and that the eIF2 α signalling-inducing transcription factor ATF3 determines this modal shift through RIPK3 induction. We also determined that ATF3-dependent RIPK3 induction plays important roles both in acute injury of steatotic liver and NASH'

Indeed, necroptosis in the severe steatosis model in Figures 1 and 2 is not analyzed. In addition, key markers of necroptosis (p-MLKL) were not analyzed in several experiments, relying only on mRNA analyses of total MLKL. The potential role of ferroptosis, which is also regulated in part by ATF3, is also not investigated. The impact of the manuscript in its current form is judged to be moderate, but to a potentially limited audience.

In response to the reviewer's suggestion, we newly measured MLKL phosphorylation (phospho-MLKL) to evaluate necroptosis, and the results have been added to the original Figs. 1 and 2. We found that the levels of phospho-MLKL increased in severe steatosis after hepatectomy, as shown in a new Fig. 1f. RIPK3-knockdown (R-KD) and ATF3-knockout (A-KO) exhibited diminished levels of phospho-MLKL, as shown in a new Fig. 2f and new Fig. 3i, respectively. These results indicated that hepatectomy increased and R-KD and A-KO decreased necroptosis in severe hepatic steatosis. In the Results, we have explained the phospho-MLKL data.

Regarding the analyses of necroptosis in the original Fig. 1, we have performed RIPK3 knockdown (R-KD) to investigate the importance of necroptosis in HFD-induced hepatic steatosis after hepatectomy. In severe steatosis, CI-CASP3-negative (CI-CASP3⁻) hepatocellular death increased, and most of this CI-CASP3⁻ death disappeared in R-KD. These results indicate that necroptosis contributes to this decrease in CI-CASP3⁻ hepatocellular death in severe hepatic steatosis after hepatectomy. In the original Fig. 2, we have shown the results of hepatocyte-specific ATF3 knockout mice (A-KO). TUNEL/CI-CASP3 double fluorescence staining revealed that ATF3 deficiency decreased CI-CASP3⁻ hepatocellular death, most of which was necroptotic, as we showed in the original Fig. 1. These findings supported the notion that ATF3 deficiency diminishes CI-CASP3⁻ necroptosis in severe steatosis after hepatectomy.

Regarding our other results related to p-MLKL and ferroptosis, we describe them in our

responses to comments 3 and 5, respectively.

References

1. Michalopoulos, G.K. Liver regeneration after partial hepatectomy: critical analysis of mechanistic dilemmas. *Am J Pathol* 176, 2-13 (2010).
2. Campana, L., Esser, H., Huch, M. & Forbes, S. Liver regeneration and inflammation: from fundamental science to clinical applications. *Nat Rev Mol Cell Biol* (2021).

Comments.

1. Other groups previously showed that loss of RIPK3 was protective against fibrosis in murine models of NASH (Gautheron et al 2014), and that RIPK3 and MLKL were increased in patients with NASH (Afonso and Rodrigues et al 2015). Similar to this manuscript, Afonso and Rodrigues et al utilized a 18 week HFD and observed increased necroptosis. The authors need to analyze the prevalence of necroptosis prior to partial hepatectomy to confirm their MCD mouse model.

We again apologise for the lack of a clear description of the NASH model induced by MCD feeding in the original Fig. 6. Because we used the methionine/choline-deficient diet (MCD) feeding NASH model as a model of chronic liver damage, as described above, we have not performed hepatectomy in the MCD-NASH model. To clearly describe the models used, we have added the following to the Results: 'Next, we examined the role of ATF3-dependent RIPK3 induction in a model of chronic steatotic liver damage and regeneration, namely, a not hepatectomised, but diet-induced NASH model¹³.'

In the original Fig. 6 (and new Fig. 8), we have shown the prevalence of necroptosis prior to hepatectomy in the MCD-NASH model, using analyses of RIPK3 phosphorylation (phospho-RIPK3) and R-KD. Additionally, in response to the reviewer's suggestion, we measured phospho-MLKL levels in the new. Fig. 8i. MCD feeding increased phospho-RIPK3 and phospho-MLKL levels, and R-KD decreased their phosphorylation. Although we could not evaluate cell death using TUNEL/Ci-CASP3 double fluorescence staining due to its low sensitivity, R-KD markedly decreased hepatocellular death on TUNEL staining. Given that this decline in cell death with R-KD should involve necroptosis, we conclude that necroptosis plays an important role in the pathogenesis of MCD-induced NASH.

Previous papers (Gautheron et al. 2014, Afonso et al. 2015) used whole-body knockout of RIPK3 to reveal the importance of NASH induced by MCD. We have added Afonso et al. (2015) as a new reference (#17). In the original manuscript, we performed RIPK3 knockdown using the intravenous injection of short interfering RNA (siRNA) with a cationic lipid reagent. In response to the reviewer's suggestion, we confirmed the efficacy and specificity of this

knockdown method, as described below and shown in the new Supplementary Fig. 2a–e. This method achieved high efficacy and moderate specificity of knockdown in hepatocytes. Therefore, these findings not only confirmed the achievements of previous articles regarding the importance of necroptosis but also added new insights into the importance of hepatocellular necroptosis in an MCD-NASH model.

2. While the authors here use a model where a 70% hepatectomy is performed following development of steatosis, the pathological relevance, or the impact of this model to the scientific field, is not made clear, particularly as the rest of the manuscript focuses on steatosis and NASH-driven fibrosis. Furthermore, the hepatectomy model is not repeated in other murine models of steatosis included in the paper. The lack of consistency limits the impact of the conclusions.

In accordance with our apology above, we agree that the original manuscript lacked an explanation of why we used a hepatectomy model in addition to a NASH model of MCD feeding. Because the aim of this study was to clarify the regulation of hepatocellular death after acute and chronic liver damage in hepatic steatosis, we have added a description of this aim to the Introduction, Results and Discussion.

In terms of the rationale for using a hepatectomised HFD model and MCD-NASH model from the aspect of the strength of eIF2 α signalling and RIPK3/MLKL, we have added the results of a comparison between a hepatectomised HFD model and several un-hepatectomised NASH models, namely, a HFD feeding fatty liver model, high-fat high-cholesterol feeding (HFHC) NASH model and MCD-NASH model, as shown in a new Fig. 8a. eIF2 α signalling activation was strong enough to induce RIPK3/MLKL activation in the MCD-NASH model as a hepatectomised HFD model but not in the un-hepatectomised HFD-fatty liver model and HFHC-NASH model. To the Results, we have added the following explanation of these data: 'We compared eIF2 α signalling activity between NASH models of chronic liver damage and hepatectomised hepatic steatosis models of acute liver damage. While both high-fat high-cholesterol diet (HFHC) feeding for 28 weeks and MCD feeding for 6 weeks result in NASH with hepatic inflammation and fibrosis²⁸, MCD feeding evoked a sufficiently potent activation of eIF2 α signalling to induce the protein and mRNA expression of ATF3 and RIPK3, as in hepatectomised HFD-induced hepatic steatosis, but not with HFHC feeding (Fig. 8a, b). RIPK3 and MLKL phosphorylation was also increased by MCD feeding (Fig. 8a).' To the Discussion, we have added the following: 'Furthermore, 28-week HFHC feeding, which is used to create a NASH model²⁸, was associated with low expression of ATF3 and RIPK3. Under these conditions, the hepatocellular stress may be too weak to induce ATF3 and RIPK3.'

3. The authors analyzed MLKL in whole liver using qPCR, but did not show levels of total or phosphorylated MLKL. Indeed, p-MLKL is missing from several in vivo and in vitro analyses.

We measured the protein expression of MLKL and phospho-MLKL in all of the mouse models. Specifically, we have added the data concerning the hepatectomised HFD model to Fig. 1f, the hepatectomised HFD model with R-KD to Fig. 2f, the hepatectomised HFD model with A-KD to Fig. 3i, the un-hepatectomised HFD model with ATF3 overexpression to Fig. 4g, the hepatectomised HFD model with GADD34 overexpression (newly performed in response to the suggestion of Reviewer #2) to Fig. 5a, the hepatectomised HFD model with TNF α neutralisation (newly performed in response to the suggestion of both reviewers) to Fig. 6e, the comparison between the hepatectomised HFD model and several un-hepatectomised NASH models to Fig. 8a, the MCD-NASH model with R-KD to Fig. 8i and the MCD-NASH model with A-KO to Fig. 9f. We have also added explanations of these results to the Results.

We also measured phospho-RIPK3 and MLKL in isolated hepatocytes from lean and obese mice with knockdown of ATF3 or RIPK3. However, their levels were undetectable in these hepatocytes. Therefore, to the Results, we have added the following: 'Phosphorylation of MLKL and RIPK3 was not detected in these isolated hepatocytes from both lean and obese mice (Fig. 6d).'

4. Both TNF and Caspase-8 are a key regulator of necroptosis in mice. The authors should 1) analyze whether the increased necroptosis is dependent on, or independent of, caspase-8 levels, and 2) determine whether ATF3 upregulation is downstream of TNF.

1) We measured cleaved-caspase 8 (Cl-CASP8) in vivo and in vitro. We found that Cl-CASP8 had the same tendency as cleaved-caspase 3 (Cl-CASP3) in mouse models. We have shown the in vivo data of Cl-CASP8 in the respective figures showing MLKL and phospho-MLKL, which were described in our response to comment 3.

We also measured Cl-CASP8 in H4IIE hepatoma cells the death of which is induced by TNF α . Although there was no difference in Cl-CASP3 expression between ATF3-overexpressing cells with and without the RIPK3 inhibitor GSK872, this inhibitor showed a tendency for a slight increase in Cl-CASP8. However, ATF3 overexpression and GSK872 treatment affected cell viability and the rate of apoptosis, and these factors might affect the activation of CASP8 in each cell. Therefore, in these experiments, we cannot show clear evidence on whether or not Cl-CASP8 affects ATF3-dependent TNF α -induced necroptosis.

2) In response to the reviewers' suggestion, we conducted TNF α neutralisation to elucidate the role of TNF α in the ATF3-dependent induction of necroptosis, as shown in a new Fig. 5e–h. Because we clarified the role of ATF3 and TNF α in the necroptosis occurring

during the regeneration after liver damage, we express our sincere thanks to the reviewer. In particular, we performed TNF α neutralisation in a model of hepatectomised severe hepatic steatosis induced by HFD and found that ATF3 induced RIPK3 and TNF α activated RIPK3 without affecting ATF3 expression. To the Results, we have added a new paragraph with the subheading 'TNF α neutralisation prevents RIPK3 phosphorylation after hepatectomy'. To the Discussion, we have added the following: 'While ATF3 is the inducer of RIPK3, TNF α is a vital activator of RIPK3 in hepatic steatosis. Indeed, neutralisation of TNF α diminished the phosphorylation of RIPK3 but did not affect ATF3 expression. Furthermore, in isolated hepatocytes, ATF3 expression induced RIPK3 expression, but did not always induce phosphorylation of RIPK3 and MLKL. These results indicate that RIPK3 induction by ATF3 and its activation by TNF α are produced independently in hepatic steatosis.'

5. The authors should assess the potential contribution of ferroptosis in their models, particularly since ATF3 can also promote ferroptosis.

We thank the reviewer for this suggestion. To investigate ferroptosis in our models, we measured the hepatic expression of the anti-ferroptotic genes Gpx4 and Slc7a11 and the hepatic levels of the lipid peroxidation biomarker malondialdehyde (MDA). We have added the following data to the appropriate figures: from a hepatectomised HFD model to Fig. 1g&h, from a hepatectomised HFD model with A-KD to Fig. 3h&j, from an un-hepatectomised HFD model with ATF3 overexpression to Fig. 4h&i, from a hepatectomised HFD model with GADD34 overexpression (newly performed in response to the suggestion of Reviewer #2) to Supplementary Fig. 5k&l, from a hepatectomised HFD model with TNF α neutralisation (newly performed according to the suggestion of both reviewers) to Supplementary Fig. 5n&o, from MCD-NASH models to Supplementary Fig. 9g&h and from an MCD-NASH model with A-KO to Fig. 9g&h. We have also added their explanations to the Results. These data indicated that hepatectomy and MCD feeding induced ferroptosis, but that ATF3-induced necroptosis in severe hepatic steatosis had little effect on ferroptosis. To the Introduction, we have added the following: 'Ferroptosis, another mode of lytic cell death that is characterised by iron-dependent and lipid peroxidation-mediated cell death, may play a role in the pathogenesis of NASH^{4,18,19}. Malondialdehyde (MDA), a biomarker of lipid peroxidation, is increased in murine models of NASH or NASH patients^{19,20}. While hepatic glutathione peroxidase 4 (GPX4), a key protector against lipid peroxidation, is increased in murine NASH models, drug-mediated inhibition or activation of GPX4 exacerbates or ameliorates the severity of murine NASH¹⁹. However, it remains unknown what cell death modes are selected in steatotic hepatocytes and the mechanism involved.' To the Discussion, we have added the following: 'Ferroptosis, an iron-dependent type of lytic cell death, is mediated by lipid

peroxidation, and glutathione and GPX4 reduce lipid peroxides and inhibit ferroptosis⁴. ATF3 is reported to promote ferroptosis by decreasing the expression of Slc7a11, which encodes a cystine transporter and which is essential for glutathione synthesis, in a human sarcoma cell line⁴¹. In this study, we found that hepatectomy of steatotic liver or MCD feeding increased ferroptosis, as indicated by the elevated MDA levels and as reported previously¹⁹. While ferroptosis contributes somewhat to the cell death, the reduction in cell death with R-KD revealed that the dominant cell death was necroptosis under these conditions of acute and chronic hepatic steatosis damage. Furthermore, neither knockout nor overexpression of ATF3 produced any change in hepatic MDA levels in vivo, indicating that ATF3 plays only a limited role in the induction of ferroptosis in the liver of the models used in this study. A relatively short period of feeding with an MCD or choline-deficient, ethionine-supplemented diet induces ferroptosis^{19,42}, indicating that ferroptosis may play an important role in the early phase of NASH. In our results, ATF3 overexpression increased the expression of Gpx4 and Slc7a11 in addition to Tnf, while TNF α neutralisation decreased their expression. TNF α may induce the expression of these anti-ferroptotic genes in hepatocytes, in line with a previous report that TNF α protects against ferroptosis by inducing anti-ferroptotic genes in fibroblasts⁴³.

6. The use of an siRNA to reduce RIPK3 in mice, as well as the adenoviral overexpression, raises questions regarding the specificity of the results. RIPK3 has been shown to be highly expressed by cholangiocytes, as well as non-parenchymal cells in the liver, whereas overexpression could impact several hepatic cell types. This is additionally called into question, as TNF expression is reduced in MCD mice receiving siRNA targeting RIPK3, which is a canonical inducer of necroptosis.

We have examined the hepatic specificity of siRNA knockdown and found that this method of knockdown displayed moderate specificity to hepatocytes in the liver, as shown in a new Supplementary Fig. 2a–d. We have added the following to the Results: ‘To determine whether necroptosis is the dominant mode of cell death during the regeneration of severely steatotic livers, we used hepatocyte Ripk3 knockdown (R-KD) with hepatectomy. For hepatic knockdown, we used intravenous injection of short interfering RNA (siRNA) with a cationic lipid reagent. First, to examine the hepatic cellular specificity of the knockdown, we intravenously administered siGapdh. Gapdh mRNA was decreased in hepatocytes expressing Alb but not in hepatic non-parenchymal cells expressing Adgre1, Cadh5 or Des, indicating that this knockdown displayed moderate specificity for hepatocytes in the liver (Supplementary Fig. 2a–e).’

Regarding the decrease in TNF α with R-KD, we believe that this decrease was due to

the decrease in hepatic necroptosis. Although TNF α is a major inducer of necroptosis, lytic cell death, such as that of necroptosis, is also known to induce inflammation around dead cells by releasing pro-inflammatory intracellular molecules called damage-associated molecular patterns (DAMPs). Therefore, there is a vicious cycle of necroptosis and inflammation. R-KD can break this cycle, resulting in a decrease in hepatic TNF α expression. To the Discussion, we have added the following: 'Given that TNF α leads to RIPK3 activation and necroptosis induction, which increases inflammation and TNF α levels^{4,9}, ATF3-dependent induction of RIPK3 in severe hepatic steatosis can cause a vicious cycle of necroptosis and inflammation. Indeed, hepatocellular deficiency of RIPK3 or ATF3 decreased hepatic TNF α expression and ATF3 overexpression increased its expression. This vicious cycle can play an important role in the pathogenesis of the exacerbation of liver damage.'

7. In the overexpression models, a single adenoviral infection was performed. Did the expression of ATF3 persist? Was the expression preferentially in hepatocytes, or was it ubiquitous throughout the liver?

We examined the hepatic specificity of adenovirus-mediated overexpression and found that this method of overexpression enabled hepatocyte-dominant gene transfection evenly across the liver, as shown in a new Supplementary Fig. 4a–g. To the Results, we have added the following: 'We performed hepatic overexpression using intravenous injection of adenoviral vector. To examine intrahepatic cellular specificity, we used adenoviral vector encoding mCherry and found that intravenous injection of this vector enabled hepatocyte-dominant gene transfection (Supplementary Fig. 4a–e). We determined that each lobe of the liver expressed mCherry at the same level and throughout the hepatic lobe (Supplementary Fig. 4f, g).'

In response to the suggestion of Reviewer #2, we have performed adeno-associate virus (AAV)-mediated overexpression of GADD34, a phosphatase of eIF2 α . Therefore, we have also checked the hepatic specificity of AAV. AAV intravenous injection also enabled hepatocyte-dominant gene transfection ubiquitously throughout the liver, as shown in a new Supplementary Fig. 5a–g. To the Results, we have added the following: 'Because we adopted gene transfection with less liver damage for the sake of hepatectomy rather than adenovirus vector-mediated transfection, we performed hepatic overexpression via intravenous injection of a chimeric adeno-associated viral vector (AAV), AAV-DJ, which enables gene transfection in the liver⁸. To examine intrahepatic cellular specificity, we used AAV encoding green fluorescent protein (GFP) and found that intravenous injection of this vector induced ubiquitous hepatocyte-dominant gene transfection throughout the liver (Supplementary Fig. 5a–g).'

8. Recent manuscripts show that ATF3 promotes NASH (Tu et al, Hepatology 2020)? the authors need to reconcile the literature with their findings that ATF3 loss did not seem to impact steatosis development in Extended Data Figure 2.

Tu et al. (Hepatology, 2020) and Kim et al. (J Hepatol, 2017) reported that ATF3 knockdown decreases hepatic lipid accumulation in cultured hepatocytes with palmitate treatment and in the liver of Zucker diabetic fatty (ZDF) rat. We have added these articles as new references. In this study, ATF3 deficiency produced no change in hepatic triglyceride levels in HFD-fed mice. Because the ZDF rat has much more severe obesity and hyperglycemia compared with HFD-fed mice, this discrepancy in the hepatic triglyceride levels may be due to the difference in background metabolic phenotypes. To the Discussion, we have added the following: 'Previous work indicated that hepatic ATF3 knockdown decreased the levels of hepatic lipogenic genes and triglycerides in cultured hepatocytes treated with palmitate and in the liver of Zucker diabetic fatty (ZDF) rat^{39,40}. Here, ATF3 deficiency produced no change in hepatic triglyceride levels in pre-hepatectomised HFD-fed mice, probably due to their milder phenotype of obesity and hyperglycemia compared with ZDF rat. However, given that ATF3 increases hepatic lipogenesis and lipid accumulation^{39,40}, ATF3 may exacerbate NASH through a synergistic effect on lipogenesis and hepatocellular death in the presence of severe obesity and hyperglycemia.'

9. The results section would greatly benefit from additional introduction of the methods/models used, particularly in vivo, in order to improve the ease of understanding and clarity for readers.

As the reviewer pointed out, the original manuscript is hard to understand due to the use of multiple models. Therefore, we have revised the Results section, particularly the subheadings of each paragraph, to improve reader understanding of the model addressed in each paragraph. We have also added a further explanation and conclusion to each paragraph.

Reviewer #2 (Remarks to the Author):

The manuscript from Inaba et al. titled 'ATF3 switches cell death from apoptosis to necroptosis in hepatic steatosis' demonstrated that increased hepatic steatosis severity shifts the hepatocellular death mode from apoptosis to necroptosis, aggravating the liver damage. In vivo, ATF3 regulates the expression of RIPK3. Under severe hepatic steatosis conditions, hepatic ATF3-deficient or -overexpressing mice display decreased or increased necroptosis,

respectively. In vitro studies implicate that ATF3 increases RIPK3 expression by directly binding to its gene promoter leading to necroptosis. In mice and patients with NASH disease severity appears to associate with ATF3 and RIPK3 expression in hepatocytes. This study points out a novel function of ATF3 related to necroptosis in liver steatosis injury.

However, some issues remain to be considered:

1. The authors indicated that ATF3 is induced in hepatocytes during liver steatosis injury. What is exactly the mechanism associated with this increased expression?

The eIF2 α and JNK-c-Jun signalling pathways induce ATF3 expression^{3,4} To investigate the role of eIF2 α signalling in the induction of ATF3, we have performed hepatic overexpression of GADD34, a phosphatase of eIF2 α , in hepatectomised HFD mice, as shown in a new Fig. 5a–d. GADD34 overexpression decreased eIF2 α phosphorylation and ATF3 expression. However, TNF α neutralisation, which we also newly performed, decreased c-Jun phosphorylation, but not ATF3 expression, as shown in a new Fig. 5e&f. These results indicated that eIF2 α signalling plays an important role in ATF3 induction in hepatectomised HFD mice. To the Results, we have added a new paragraph concerning a hepatectomised HFD model with GADD34 overexpression, entitled ‘**Dephosphorylation of eIF2 α decreases ATF3/RIPK3 induction after hepatectomy**’. We have also added another paragraph on a hepatectomised HFD model with TNF α neutralisation to the Results, entitled ‘**TNF α neutralisation prevents RIPK3 phosphorylation after hepatectomy**’. To the Discussion, we have added the following: ‘**In this study, we found that hepatic ATF3 induction depends on eIF2 α signalling, which is a stress response pathway induced by various intracellular stresses. Indeed, inhibition of the eIF2 α signalling pathway by overexpression of GADD34, a phosphatase of eIF2 α , decreased the expression of ATF3 and RIPK3 after resection of severe hepatic steatosis. In addition to eIF2 α signalling, JNK-c-Jun signalling also induces ATF3 expression^{37,38}. TNF α neutralisation decreased c-Jun phosphorylation, but not ATF3 expression, after hepatectomy in HFD-induced severe hepatic steatosis, indicating that JNK-c-Jun signalling may have an insignificant effect on ATF3 induction in this model.**’

References

3. Fu, L. & Kilberg, M.S. Elevated cJUN expression and an ATF/CRE site within the ATF3 promoter contribute to activation of ATF3 transcription by the amino acid response. *Physiol Genomics* 45, 127-137 (2013).
4. Hayner, J.N., Shan, J. & Kilberg, M.S. Regulation of the ATF3 gene by a single promoter in response to amino acid availability and endoplasmic reticulum stress in human primary hepatocytes and hepatoma cells. *Biochim Biophys Acta Gene Regul Mech* 1861, 72-

79 (2018).

2. What is the potential involvement of specific pathways governing tissue inflammation on the role of ATF3 in liver steatosis injury? Does RNA expression data suggest that inflammatory responses are one of the key pathways involved?

In response to the reviewer's suggestion, we have examined TNF α neutralisation in severe hepatic steatosis after hepatectomy and determined that ATF3 induces RIPK3 expression and that TNF α activates this RIPK3 induction. We have added the TNF α neutralisation data to a new Fig. 5e–h and a new Supplementary Fig. 5m–o. To the Results, we have added a new paragraph entitled, 'TNF α neutralisation prevents RIPK3 phosphorylation after hepatectomy'. To the Discussion, we have added the following: 'While ATF3 is the inducer of RIPK3, TNF α is a vital activator of RIPK3 in hepatic steatosis. Indeed, neutralisation of TNF α diminished the phosphorylation of RIPK3 but did not affect ATF3 expression. Furthermore, in isolated hepatocytes, ATF3 expression induced RIPK3 expression, but did not always induce phosphorylation of RIPK3 and MLKL. These results indicate that RIPK3 induction by ATF3 and its activation by TNF α are produced independently in hepatic steatosis.' and 'Given that TNF α leads to RIPK3 activation and necroptosis induction, which increases inflammation and TNF α levels^{4,9}, ATF3-dependent induction of RIPK3 in severe hepatic steatosis can cause a vicious cycle of necroptosis and inflammation. Indeed, hepatocellular deficiency of RIPK3 or ATF3 decreased hepatic TNF α expression and ATF3 overexpression increased its expression. This vicious cycle can play an important role in the pathogenesis of the exacerbation of liver damage.'

3. In human studies, what are the mRNA levels of ATF3 and RIPK3? The human study will need to be further strengthened by additional functional analysis between ATF3/RIPK3 levels and liver injury outcomes (e.g. ALT, AST). Moreover, what are the levels of RIPK3 phosphorylation in low and high-NAS score patients?

We agree that the mRNA levels of ATF3 and RIPK3 would help to elucidate the relationship between ATF3 and RIPK3. However, we unfortunately do not have mRNA from NASH patients.

We have performed additional functional analyses comparing ATF3/RIPK3 levels and plasma aminotransferase levels. Only phospho-RIPK3 was correlated with plasma AST levels, as shown in a new Fig. 10d and Supplementary Fig. 10b. Given that ATF3 and RIPK3 enable hepatocytes to acquire a new ability to undergo necroptosis and TNF α triggers necroptosis via phospho-RIPK3, this correlation seems to be reasonable.

We also performed immunostaining of phospho-RIPK3. Phospho-RIPK3 levels were

correlated with RIPK3 and ATF3 levels (Fig. 10a), plasma aminotransferase levels (Fig. 10d) and fibrosis stage (Fig. 10i), but not with NAS, steatosis, ballooning and inflammation (Fig. 10e–h). To the paragraph named ‘**Hepatic ATF3 and RIPK3 expression in NASH patients**’, we have added an explanation of these data.

4. In the analysis of the gene expression of necroptosis-related molecules, expression of the Mkl and Ripk1 genes was not significantly increased during the regeneration process of severely steatotic livers. Would this contradict the increased levels of necroptosis, which is triggered by the sequential phosphorylation of RIPK3 and MLKL that disrupts the cell membrane? What additional mechanism governs RIPK3-driven necroptosis?

Thanks to the reviewers’ suggestions, we have now more clearly elucidated the mechanism of the ATF3-dependent induction of necroptosis in this revised version of the manuscript. Indeed, we measured the protein expression of MLKL and phospho-MLKL in all of the mouse models. Specifically, we have added the data concerning the hepatectomised HFD model to Fig. 1f, the hepatectomised HFD model with R-KD to Fig. 2f, the hepatectomised HFD model with A-KD to Fig. 3i, the un-hepatectomised HFD model with ATF3 overexpression to Fig. 4g, the hepatectomised HFD model with GADD34 overexpression (newly performed in response to the suggestion of Reviewer #2) to Fig. 5a, the hepatectomised HFD model with TNF α neutralisation (newly performed in response to the suggestion of both reviewers) to Fig. 6e, the comparison between the hepatectomised HFD model and several un-hepatectomised NASH models to Fig. 8a, the MCD-NASH model with R-KD to Fig. 8i and the MCD-NASH model with A-KO to Fig. 9f. We have also added explanations of these results to the Results.

We also performed TNF α neutralisation in a hepatectomised HFD model, as described above in our response to comment 2. Through these analyses, we found that it is TNF α that phosphorylates hepatocellular RIPK3 and MLKL.

Because MLKL is expressed in hepatocytes under basal conditions, but not RIPK3, as previously reported^{5,6}, the presence of RIPK3 determines whether hepatocytes die via necroptosis or not. Indeed, in Fig. 2, we showed that R-KD decreased non-apoptotic hepatocellular death after hepatectomy in severe hepatic steatosis, indicating that TNF α should induce necroptosis in the presence of RIPK3, but apoptosis in the absence of RIPK3. Therefore, in conclusion, RIPK3 and its inducer ATF3 enable hepatocytes acquire a new ability to undergo necroptosis and TNF α triggers necroptosis via phosphorylation of RIPK3.

References

5. Dara, L., Liu, Z.X. & Kaplowitz, N. Questions and controversies: the role of

necroptosis in liver disease. *Cell Death Discov* 2, 16089 (2016).

6. Gautheron, J., Gores, G.J. & Rodrigues, C.M.P. Lytic cell death in metabolic liver disease. *J Hepatol* 73, 394-408 (2020).

REVIEWERS' COMMENTS

Reviewer #1 (Remarks to the Author):

The authors have addressed the critiques and have increased the clarity of the manuscript.

Reviewer #2 (Remarks to the Author):

This study provides interesting and novel findings showing that hepatocellular death switches from apoptosis to necroptosis via ATF3-dependent RIPK3, both in acute liver damage in hepatic steatosis and in chronic liver injury and also in NASH patients. The authors have adequately addressed my concerns.

Response to Reviewers

We would like to thank the reviewers and editorial staff for their helpful comments and for reviewing our manuscript. We have included responses to each reviewer comment below, as well as our responses to comments from the editorial staff in the previously submitted Nature Reporting Summary.

Response to Reviewer #1

Comment: The authors have addressed the critiques and have increased the clarity of the manuscript.

Response: We thank the reviewer for this comment and for their previous advice, which has helped us to improve our manuscript.

Response to Reviewer #2

Comment: This study provides interesting and novel findings showing that hepatocellular death switches from apoptosis to necroptosis via ATF3-dependent RIPK3, both in acute liver damage in hepatic steatosis and in chronic liver injury and also in NASH patients. The authors have adequately addressed my concerns.

Response: We thank the reviewer for this comment and for their previous advice, which has helped us to improve our manuscript.